# Ligustilide: A Phytochemical with Potential in Combating Cancer Development and Progression—A Comprehensive and Critical Review

**DOI:** 10.3390/ph19010036

**Published:** 2025-12-23

**Authors:** Victória Dogani Rodrigues, Mayara Longui Cabrini, Eliana de Souza Bastos Mazuqueli Pereira, Manuela dos Santos Bueno, Virgínia Maria Cavallari Strozze Catharin, Jesselina Francisco dos Santos Haber, Rachel Gomes Eleutério, Lidiane Indiani, Vitor Cavallari Strozze Catharin, Raquel Cristina Ferraroni Sanches, Flávia Cristina Castilho Carácio, Tereza Lais Menegucci Zutin, Vitor Engrácia Valenti, Sandra Maria Barbalho, Lucas Fornari Laurindo

**Affiliations:** 1Department of Biochemistry and Pharmacology, School of Medicine, Faculdade de Medicina de Marília (FAMEMA), Marília 17519-030, São Paulo, Brazil; 2Department of Physiotherapy, School of Physiotherapy, Universidade de Marília (UNIMAR), Marília 17525-902, São Paulo, Brazil; 3Graduate Program in Structural and Functional Interactions in Rehabilitation, School of Medicine, Universidade de Marília (UNIMAR), Marília 17525-902, São Paulo, Brazil; 4Department of Biochemistry and Pharmacology, School of Medicine, Universidade de Marília (UNIMAR), Marília 17525-902, São Paulo, Brazil; 5Department of Odontology, School of Dentistry, Universidade de Marília (UNIMAR), Marília 17525-902, São Paulo, Brazil; 6Faculty of Philosophy and Sciences, Universidade Estadual Paulista (UNESP), Marília 17525-900, São Paulo, Brazil; 7Laboratory for Systematic Investigations of Diseases, Department of Biochemistry and Pharmacology, School of Medicine, Universidade de Marília (UNIMAR), Marília 17525-902, São Paulo, Brazil; 8Division of Cellular Growth, Hemodynamic, and Homeostasis Disorders, Graduate Program in Medical Sciences, Faculdade de Medicina, Universidade de São Paulo (USP), São Paulo 01246-903, São Paulo, Brazil

**Keywords:** Ligustilide, (Z)-Ligustilide, cancer, metastasis, cancer development, cancer progression, malignancies

## Abstract

Cancer remains one of the leading global health challenges, with increasing resistance to conventional therapies hindering treatment efficacy. Ligustilide, a bioactive compound derived from *Ligusticum chuanxiong*, has garnered attention for its multifaceted pharmacological properties, including anti-inflammatory, neuroprotective, and anticancer effects. This review comprehensively examines Ligustilide and its isomer, (Z)-Ligustilide, focusing on their anticancer potential across various cancer types. Ligustilide exerts its therapeutic effects through multiple mechanisms, including inhibition of cell proliferation, induction of apoptosis, and modulation of autophagy. Additionally, (Z)-Ligustilide has been shown to enhance drug sensitivity and modulate epigenetic regulation, providing a novel approach to overcoming chemoresistance. Despite promising preclinical results, the precise molecular mechanisms, pharmacokinetics, and bioavailability of Ligustilide remain under investigation. Future research should focus on optimizing its therapeutic applications, exploring its synergy with other chemotherapeutic agents, and assessing its potential in personalized cancer therapies. This review offers an in-depth analysis of Ligustilide’s anticancer mechanisms, its role in overcoming drug resistance, and its potential as a novel therapeutic strategy in cancer treatment.

## 1. Introduction

Cancer is a global health concern. It is the second leading cause of mortality in the United States and the top cause of death for people under 85 years old [1,2]. Tumorigenesis is a multi-stage process that begins with an oncogenic mutation in a single somatic cell, which confers a growth advantage to the cell. In this scenario, more genetic and epigenetic modifications accumulate over time, leading to invasive and heterogeneous tumors. These mutations promote clonal expansion and cancer progression. The significant risk factors associated with these conditions are environmental stressors and aging, which disrupt cellular balance, increasing cancer risk by creating a complex ecosystem [3,4]. This ecosystem, known as the tumor microenvironment, is where cancer cells are surrounded by a diverse mix of non-malignant cells, including immune cells, fibroblasts, endothelial cells, and others, embedded in a remodeled, vascularized extracellular matrix. The components of the tumor microenvironment differ across tumor locations, stages, characteristics, and patient-specific factors. These are both supporters and suppressors of the tumor growth [5,6].

In this scenario, drug resistance is a significant obstacle to effective cancer treatment. Sensitive cancers can become resistant and more aggressive over time, and resistance limits the success of even targeted therapies, despite recent approvals. Combining chemotherapeutic agents or modifying dosing strategies shows promise in reducing tumor growth and regrowth. However, targeting both intrinsic (genetic mutations, heterogeneity, and survival pathways) and extrinsic (tumor microenvironment components) forms of resistance in cancerous cells remains critical [7,8]. As a result, there is a continued need for new treatments that more precisely target cancer cells, enhance the impact of current therapies, and minimize side effects. In this sense, bioactive compounds and phytochemicals derived from medicinal plants and functional foods are gaining prominence [9,10].

Ligustilide is a bioactive component derived from the medicinal plant *Ligusticum chuanxiong*. This compound has multiple health benefits and has been studied for its potential clinical applications in combating primarily inflammatory and oxidative diseases [11]. It is considered a volatile component from *Umbelliferae* plants, notably originating in *L. chuanxiong*’s rhizome. However, it can also be encountered in *Radix Angelicae sinensis*, which is widely used in Chinese medicine. Ligustilide’s chemical structure name is 3-butylidene-4,5-dihydro-1(3H)-isobenzofuranone. Its molecular formula comprises C_12_H_14_O_2_. The phytochemical exists in two isomeric forms: (Z)-Ligustilide and (E)-Ligustilide. The Z-configuration is dominant in terms of spatial conformation, is more stable than the E-isomer, and is significantly more abundant. Ligustilide’s pharmacological activities include anti-atherosclerotic, analgesic, neuroprotective, cardioprotective, and ultimately, anticancer effects [12]. Figure 1 depicts the structure of Ligustilide based on Chen et al.’s article [13].

The anticancer activities of Ligustilide span a variety of effects. Ligustilide has been shown to inhibit cancer cell proliferation, migration, and invasion by modulating key molecular pathways, including the PI3K/Akt signaling pathway [14]. It has also limited cancer growth by inducing endoplasmic reticulum stress-induced mitochondrial dysfunction and autophagy [15]. Furthermore, it has induced ferroptosis in malignancies primarily by modulating the Nrf2/HO-1 pathway [16]. Other effects include, but are not limited to, reducing cancer’s resistance to chemotherapeutic agents, such as cisplatin, in various cancer types [17]. Together, these findings highlight Ligustilide as a promising multitargeted therapeutic agent with the potential to enhance cancer treatment efficacy and overcome drug resistance, both crucial concerns in malignancy treatment.

Recently, Liu et al. [18] attempted to assess the anticancer potential of Ligustilide. However, their approach is limited for several reasons. Firstly, their paper comprised multiple phytochemicals with anticancer potential. Therefore, their analysis lacked specificity. Additionally, their approach focused on the power of natural products to target cancer-related macrophages. Although the study cited Ligustilide as an anticancer agent, its analysis lacked a comprehensive overview of the phytochemical’s potential against various, differing cancer types. As mentioned earlier, the literature on Ligustilide’s anticancer potential remains limited, as no comprehensive review has examined its full potential across various cancer types.

To address these limitations, our comprehensive and critical review provides a focused, thorough analysis of Ligustilide’s anticancer potential. Unlike previous studies that grouped Ligustilide with other phytochemicals or discussed it only within the narrow context of cancer-associated macrophages, our work offers an in-depth evaluation of Ligustilide as a standalone compound. Based on the existing literature, we systematically explore Ligustilide’s mechanisms of action, effects across different cancer types, and therapeutic relevance, providing a more precise understanding of its role as a promising candidate in cancer therapy.

## 2. Ligustilide: Unveiling Its Biosynthesis, Physicochemical Properties, and Pharmacokinetics

Ligustilides, including (E)-Ligustilide and (Z)-Ligustilide, are volatile components commonly analyzed by GC-flame ionization detection or GC-mass spectrometry in herbs and materials containing volatile oils. Ligustilide can be easily isomerized at high temperatures, and its inherent instability hinders precise quantification [19]. The content of Ligustilide varies with herb shape, cultivating areas, and plant species. Li et al. [20] evaluated the content of (Z)-Ligustilide and (E)-Ligustilide among danggui samples. They found that the content of Ligustilide in *A. sinensis* ranged from 5.63 to 24.53 mg/g with a mean of 11.02 mg/g, with significant differences between cultivated areas in China, ranging from 13.90 mg/g in Yannan, 12.51 mg/g in Sichuan, and 10.04 mg/g in Gansu. They also found that Ligustilide content was larger in the small main root, long rootlet, and perfumed samples. Other species showed dramatically different levels, including 1.00 mg/g in *Angelica acutiloba* and *Angelica acutiloba var. sugiyamae*, and 2.78 mg/g in lovage root, *Levisticum officinale*. These findings highlight the significant variability in Ligustilide content influenced by species, plant part, and geographical origin, underscoring the importance of standardized sourcing and analytical methods in pharmacokinetic and pharmacological studies.

### 2.1. Biosynthesis of Ligustilide: Pathways and Regulatory Mechanisms

Phthalides are key active constituents of volatile oil plants. These compounds are widely found in species of the *Umbelliferae* family, including *Angelica sinensis* and *L. chuanxiong*. Although the complete biosynthetic pathway for synthesizing Ligustilide, including the interconversion mechanisms between different phthalides, has not been fully understood, some authors assert that the expression of specific enzymes in the phthalide biosynthetic pathway is a key factor. For phthalides, biosynthesis involves the structural determination of mycophenolic acid, which is a phthalide fragment derived from the polyketide pathway [21]. Mycophenolic acid is constituted by a phthalide fragment and a terpene fragment, derived from the polyketide and the isoprenoid pathways. The molecule is produced by the involvement of key enzymes, including polyketide synthases, starter unit acyl carrier protein transacylase, β-ketoacylsynthase, acyltransferase, and methyltransferase. The presence of methoxy and methyl groups in the benzene ring of the mycophenolic acid structure has also been further demonstrated. The backbone synthesis of mycophenolic acid also involves acyl carrier proteins, which are responsible for the production of the template and the production of the mpaC, which assembles as the phthalide fragment of the acid in a “gene cluster” of the *Penicillium brevicompactum* species. Research has also shown that the alkylphthalides in the formation of Ligustilide in *L. officinale* have polyketide precursors. [1′-^14^C]-4,6-dihydroxy-2,3-dimethylbenzoic acid is a precursor of the mycophenolic acid [22].

While the complete biosynthetic pathway of Ligustilide remains not fully elucidated, current evidence indicates that it originates from polyketide-derived precursors, with key enzymatic steps involving polyketide synthases and associated modifying enzymes. The formation of the phthalide structure, as seen in related compounds such as mycophenolic acid, suggests that structural and regulatory genes, organized into gene clusters, play a central role in Ligustilide biosynthesis. These findings underscore the importance of enzyme expression and precursor availability in modulating Ligustilide production, laying the groundwork for future metabolic engineering and synthetic biology approaches to enhance its yield in medicinal plants.

### 2.2. Physicochemical Properties of Ligustilide: Structural Characteristics and Stability

Ligustilide is a lipophilic phthalide derivative with a molecular formula of C_12_H_14_O_2_ and a molecular weight of 190.24 g/mol, as computed by PubChem [23]. Its partition coefficient (XLogP3-AA) is 2.7, indicating moderate hydrophobicity, which may influence its bioavailability and membrane permeability. The molecule lacks hydrogen bond donors but has two hydrogen bond acceptors, resulting in a relatively low topological polar surface area of 26.3 Å^2^. This suggests limited aqueous solubility but good potential for passive diffusion across biological membranes. Ligustilide contains two rotatable bonds, reflecting moderate molecular flexibility. Its exact and monoisotopic mass are 190.099379685 Da, indicating high precision in mass spectrometric identification. The compound has 14 heavy atoms, a complexity score of 345, and a formal charge of zero, consistent with its neutral, non-ionic nature under physiological conditions. Stereochemically, Ligustilide has one defined bond stereocenter and no undefined stereocenters, suggesting a relatively simple stereochemical profile. It also contains one covalently bonded unit and has been canonicalized in the PubChem database.

The physicochemical profile of Ligustilide indicates a moderately lipophilic molecule with favorable membrane permeability characteristics, including a low topological polar surface area and moderate hydrophobicity. While these properties support its potential for passive diffusion, the compound’s limited aqueous solubility may hinder its bioavailability. Additionally, its structural simplicity and defined stereochemistry make it amenable to analytical identification and possible chemical modification. Ligustilide’s physicochemical attributes suggest a promising drug-like profile, though formulation strategies may be necessary to address its solubility and stability limitations.

### 2.3. Pharmacokinetics of Ligustilide: Evaluating the Phytochemical’s Absorption, Distribution, Metabolism, and Toxicity

Ligustilide has been studied for its pharmacokinetic profile in rats. In these animals, the oral bioavailability of the phytochemical was estimated to be 2.6% at the 500 mg/kg dose monitored at 284 nm. After intravenous (15.6 and 14.9 mg/kg), intraperitoneal (26 and 52 mg/kg), and oral (500 mg/kg) administrations of Ligustilide, the encountered t½ (h) values were 0.31 ± 0.12 and 0.22 ± 0.07 (15.6 and 14.9 mg/kg i.v., respectively), 0.36 ± 0.05 and 0.44 ± 0.08 (26 and 52 mg/kg i.p., respectively), and 3.43 ± 1.01 (500 mg/kg p.o.). C_max_ (mg/L) values were 13.19 ± 0.84 and 6.93 ± 0.60 (15.6 and 14.9 mg/kg i.v., respectively), 7.48 ± 1.10 and 20.75 ± 2.55 (26 and 52 mg/kg i.p., respectively), and 0.66 ± 0.23 (500 mg/kg p.o.). T_max_ (h) values were 0.05 ± 0.02 and 0.08 ± 0.01 (26 and 52 mg/kg i.p., respectively) and 0.36 ± 0.19 (500 mg/kg p.o.). Ligustilide has low oral bioavailability (2.6%) due to first-pass liver metabolism. Ligustilide presents seven primary metabolites, but only three can be identified unequivocally: butylidenephthalide, senkyunolide I, and senkyunolide H after i.v. After Ligustilide administration, the compound is distributed extensively (Vd/F, 3.76 ± 1.23 L/kg) and rapidly eliminated (t½, 0.31 ± 0.12 h). The clearance after intravenous administration of Ligustilide under Chuanxiong extract was significantly higher than that of pure Ligustilide [24,25]. Ligustilide was rapidly metabolized in both rat and human hepatocyte incubations, along with high intrinsic clearance. The half-lives of Ligustilide were 8.0 and 15.0 min in rat and human incubations, respectively. Most of the parent (>90%) was transformed into metabolites, with senkyunolide I as the major metabolite, accounting for 42% in rat and 70% in human hepatocyte incubations, respectively. Researchers have reported that the metabolism of Ligustilide involves multiple pathways, including epoxidation, epoxide hydrolysis, aromatization, hydroxylation, and glutathionylation [26]. Due to the multi-conjugated, unstable structure of Ligustilide, the compound has poor drug-forming properties, making it difficult to study its toxicity in living organisms. Zhang et al. developed a highly stable, colorless needle crystal, LIGc, by structural modification of Ligustilide to assess its tissue distribution and, most importantly, evaluate the safety of the derivative. The results demonstrated that LIGc was rapidly absorbed and eliminated after intravenous and oral administration, with C_max_ of 6.42 ± 1.65 mg/L at 20 mg/kg and C_max_ of 9.89 ± 1.62 mg/L at 90 mg/kg, respectively. Data for oral administration also comprised a T_max_ of 0.5 h and a t½z of approximately 2.5 h. Mice treated with 5.0 g/kg of the Ligustilide derivative had no histopathological changes observed in the main organs. However, it is worth noting that the Ligustilide derivative had higher oral bioavailability than Ligustilide [27].

The toxicological profile of Ligustilide remains only partially defined, as most available data arise from studies using *Angelica sinensis* and *Ligusticum sinense* extracts or oils in which Ligustilide is a significant constituent rather than an isolated compound. Existing preclinical observations suggest that it is generally non-toxic at experimentally relevant doses [28]. However, formal maximum tolerated dose determinations, dose–range finding studies, and comprehensive assessments of hepatic, renal, cardiovascular, or neurological toxicity have not been conducted for purified Ligustilide. Ligustilide-rich essential oil from *Ligusticum chuanxiong* showed low acute toxicity in mice, with oral and intraperitoneal LD_50_ values thousands of times higher than clinical doses. Mild skin irritation occurred only at high external doses (over 200× the clinical amount) on rabbit skin, and no skin sensitization was observed on guinea pig skin [29]. Likewise, no standardized genotoxicity assays are currently available, and potential cardiotoxicity has not been examined beyond limited mechanistic experiments. Although Ligustilide may influence metabolic pathways, a systematic evaluation of its potential for drug–drug interactions, including CYP450 or transporter involvement, is lacking.

Given these gaps, a structured preclinical program is needed to ensure a reliable safety foundation for further development. Essential components would include basic pharmacokinetic and metabolic characterization, formal acute and subacute toxicity studies with MTD and NOAEL determination, focused safety pharmacology addressing cardiovascular, respiratory, and central nervous system effects, standard in vitro and in vivo genotoxicity assays, and subchronic toxicity studies in at least one rodent and one non-rodent species. If therapeutic use ultimately warrants, reproductive and developmental toxicity studies and drug–drug interaction profiling should also be undertaken. Overall, while preliminary evidence suggests a relatively benign safety profile, the absence of systematic toxicological evaluation remains a significant limitation and warrants dedicated investigation.

Overall, the pharmacokinetic profile of Ligustilide demonstrates rapid absorption and elimination, with low oral bioavailability primarily due to extensive first-pass metabolism and its structural instability. Although Ligustilide is extensively distributed and rapidly cleared following intravenous administration, its poor drug-like properties and complex metabolic pathways limit its therapeutic potential in its native form. The identification of key metabolites, such as senkyunolide I, and the development of more stable derivatives, such as LIGc, offer promising avenues to overcome these limitations. These findings highlight the need for further structural optimization and formulation strategies to improve Ligustilide’s pharmacokinetic characteristics and safety profile for clinical application.

### 2.4. Additional Considerations: Metabolism, Plasma Stability, Tissue Distribution, and Strategies to Improve Bioavailability

Despite the increasing number of pharmacokinetic studies on Ligustilide, several key aspects remain insufficiently characterized. Metabolic enzyme involvement has been partially elucidated: available preclinical data indicate that Ligustilide undergoes rapid metabolism. CYP3A4, CYP2C9, and CYP1A2 may mediate this [26]. Its inherent structural instability and very low bioavailability—with degradation half-lives of only minutes to a few hours in the rat—contribute substantially to its low systemic exposure and complicate accurate pharmacokinetic characterization [25,30]. The metabolite profile of Ligustilide includes at least seven major structures, with senkyunolide I, senkyunolide H, and butylidenephthalide consistently identified across studies [24].

Studies show that Ligustilide is highly lipophilic and has broad tissue distribution, including the ability to cross the blood–brain barrier, supporting experimental observations of neuroprotective effects. Its poor oral bioavailability is due to its poor stability and extensive first-pass hepatic metabolism. Ligustilide undergoes oxidative and reductive biotransformation, with the kidneys and the biliary system being the major contributors to its elimination from the body. Ligustilide degrades upon exposure to heat, light, or oxygen [31,32].

Although distribution to solid tumors has not been systematically quantified, Ligustilide’s rapid clearance may likely limit tumor exposure. The development of more stable analogs, such as LIGc, demonstrates improved oral availability and may allow more reliable assessment of tissue penetration, but comprehensive ADME characterization is still needed [33,34].

Given their inherently low oral bioavailability and rapid clearance, several formulation strategies have been proposed to improve the therapeutic viability of Ligustilide and (Z)-Ligustilide. These include nanoemulsions [35], nanoparticles [36], liposomes [37], and cyclodextrin inclusion complexes [38]. Nano-encapsulation approaches have demonstrated improved stability, prolonged circulation time (increased half-life), and increased cellular uptake [36]. Nevertheless, these strategies also pose risks, as increasing systemic exposure may exacerbate off-target effects.

Overall, integrating enzyme-specific metabolism data, plasma stability issues, tissue distribution patterns, and formulation-based enhancement strategies provides a more complete pharmacokinetic profile. These considerations underscore that while Ligustilide has promising bioactivity, its therapeutic development will require chemical stabilization, metabolic modulation, or advanced delivery systems to overcome intrinsic biopharmaceutical limitations.

## 3. Anti-Inflammatory and Antioxidant Pharmacodynamics of Ligustilide: Mechanisms of Action and Therapeutic Potential

The following lines briefly describe Ligustilide’s mechanisms of action against oxidative stress and inflammation, going beyond its specific cancer-targeting to fully demonstrate this phytochemical’s health and pharmacological potential. It is interesting to note that, although antioxidant and anti-inflammatory effects are related to cancer prevention and intervention, within the following sections, all relevant studies on Ligustilide specifically targeting cancer will be appropriately assessed as the primary interest of the present manuscript.

In terms of inflammation, Ligustilide was found to target EGR1, inhibiting the EGR1-ADAM17-TNF-α pathway to alleviate macrophage-mediated intestinal inflammation and restore the gut barrier in a colitis animal model [39]. On the other hand, Wang et al. reported the pharmacological potential of (Z)-Ligustilide in alleviating intervertebral disk degeneration in rats by suppressing nucleus pulposus cell pyroptosis via the ATG5/NLRP3 axis [40]. In nucleus pulposus cells under IL-1β stimulation, Ligustilide also inhibits the expression of inflammatory mediators (iNOS and COX-2) and reduces the production of inflammatory cytokines (TNF-α and IL-6) by inhibiting the NF-κB signaling pathway [41]. In a cardiotoxicity model of glucolipotoxicity-induced cardiomyocyte dysfunction, (Z)-Ligustilide was found to significantly protect against inflammatory injury by modulating the AMPK/GSK-3β/Nrf2 signaling pathway. This also protects against oxidative insult and fibrosis [42]. To counteract inflammation, Ligustilide also interferes with PPARγ signaling, inhibiting PPARγ-mediated inflammation [43]. By regulating mitochondrial-related inflammation in SAMP8 mice, Ligustilide improved aging-induced memory deficit [44]. Ligustilide also has immunomodulatory effects in counteracting the inflammatory response. It significantly suppresses AP-1 and the Akt/NF-κB signaling pathway, inhibits CD137 signaling, and suppresses atherosclerosis-related inflammation [45]. To prevent atheroma formation, Ligustilide also attenuates vascular inflammation by inhibiting the activation of the NF-κB signaling pathway by TNF-α [46]. In another study, (Z)-Ligustilide inhibited inflammation by suppressing NF-κB and upregulating Nrf2/HO-1 in ultraviolet B-induced human keratinocytes [47].

In summary, Ligustilide exerts potent anti-inflammatory effects by modulating several key signaling pathways, including NF-κB, PPARγ, AP-1, and AMPK/GSK-3β/Nrf2. These mechanisms contribute to its protective roles in various inflammation-related conditions, such as intestinal inflammation, intervertebral disk degeneration, cardiovascular injury, neuroinflammation, and skin damage. Ligustilide alleviates acute and chronic inflammation and helps restore tissue homeostasis by regulating inflammatory mediators and cytokine production. These findings support its broader pharmacological potential and underscore the relevance of its anti-inflammatory properties in contexts beyond cancer, further reinforcing its value as a multifunctional phytochemical worthy of deeper investigation.

Regarding oxidative damage, Ligustilide has been demonstrated to be a potent antioxidant in several studies. Yao et al. [48] showed that Ligustilide significantly enhanced mitophagy and decreased oxidative stress-induced neuronal apoptosis in an animal model of spinal cord injury due to BNIP3-LC3 interaction. On the other hand, (Z)-Ligustilide provided neuroprotection against Parkinson’s disease in a mouse model by regulating microglial phenotypic polarization via activating the Nrf2-TrxR axis [49]. Xia et al. [50] reported that Ligustilide significantly alleviated oxidative stress during renal ischemia–reperfusion injury by maintaining Sirt3-dependent mitochondrial homeostasis. In a mouse model of Alzheimer’s disease, Ligustilide provided antioxidant effects by modulating the PKA/AKAP1 signaling pathway and improving cognitive function [37]. In another study, Ligustilide was found to modulate oxidative stress, apoptosis, and immunity in bleomycin-induced pulmonary fibrosis rats by inactivating the pathway and rebalancing Th1/Th2 immunity [51]. Regarding cardiovascular diseases and outcomes, (Z)-Ligustilide was found to potently inhibit oxidative stress in a high-fat diet-induced atherosclerosis model by activating multiple Nrf2 downstream genes [52].

In summary, Ligustilide exhibits potent antioxidant activity by modulating numerous molecular targets and pathways, including the Nrf2-TrxR axis, BNIP3-LC3-mediated mitophagy, Sirt3-dependent mitochondrial homeostasis, and the PKA/AKAP1 signaling cascade. These mechanisms contribute to its neuroprotective, renoprotective, and cardioprotective effects in various models of oxidative stress-related diseases such as spinal cord injury, Parkinson’s disease, Alzheimer’s disease, pulmonary fibrosis, and atherosclerosis. By preserving mitochondrial function, enhancing antioxidant defenses, and attenuating oxidative damage, Ligustilide plays a key role in maintaining cellular redox balance and preventing oxidative injury.

The dual anti-inflammatory and antioxidant pharmacodynamics of Ligustilide collectively underscore its therapeutic potential across a wide range of pathological conditions. Its ability to target multiple signaling pathways central to inflammation and oxidative stress—both hallmarks of chronic diseases, including cancer—positions Ligustilide as a promising multifunctional phytochemical. These findings provide a strong foundation for future investigations into its application in cancer therapeutics and broader clinical contexts where inflammation and oxidative stress are pivotal. Figure 2 summarizes the dual anti-inflammatory and antioxidant activities of Ligustilide, highlighting its primary molecular targets—NF-κB, PPARγ, AP-1, AMPK (inflammation) and Nrf2, BNIP3, Sirt3, AKAP1 (oxidative stress). The diagram also illustrates the compound’s broad biological effects, including immunomodulation, tissue homeostasis, redox balance, mitochondrial protection, and overall cytoprotection.

## 4. Ligustilide in Cancer Prevention and Intervention

This section explores the growing body of evidence surrounding Ligustilide’s anticancer potential. Recent studies have highlighted its potential to modulate critical cancer-related processes, including tumor growth, apoptosis, and metastasis. Ligustilide has emerged as a promising candidate in cancer therapy, with research suggesting its ability to target multiple molecular pathways involved in cancer progression. This review aims to summarize the latest findings on Ligustilide’s mechanisms of action, evaluate its therapeutic prospects, and consider its role in shaping future cancer treatment strategies.

### 4.1. Literature Search Methodology

A comprehensive and critical literature search was performed across reputable databases, including PubMed, Scopus, Web of Science, Embase, and Google Scholar. The rationale for conducting this comprehensive review is that bioactive compounds often exhibit anticancer effects. Notably, no published reviews have fully addressed the anticancer potential of Ligustilide, particularly given the compound’s demonstrated impacts and mechanisms of action, alongside the growing body of evidence published each year. Keywords were used to facilitate the literature review. They included terms such as “Ligustilide,” “cancer cell lines,” “animal models,” “cancer,” “cell lines,” “breast cancer,” “oral cancer,” “gastric cancer,” “lung cancer,” and “signaling pathways,” alongside biological processes like “apoptosis,” “cell proliferation,” “metastasis,” “cell death,” “cell cycle,” “PI3K,” “Nrf2,” “mTOR,” “NF-κB,” “cancer prevention,” “chemoprevention,” “carcinogenesis prevention,” “DNA damage protection,” and “anti-inflammatory mechanisms.” Due to the absence of clinical trials on Ligustilide, the inclusion criteria focused on in vitro and in vivo studies investigating its anticancer effects. The selected studies analyzed the impact of Ligustilide on various cancer cell lines and animal models across a range of cancer types. Key outcomes of interest included cell viability, apoptosis, migration, treatment options and durations, administration routes, and the modulation of molecular pathways relevant to cancer progression. Exclusion criteria included reviews, meta-analyses, poster presentations, editorials, communications, letters to the editor, non-experimental papers, and studies that did not involve Ligustilide as an intervention or were not based on cancer cell lines or relevant animal models. No time restrictions were placed on the literature search to ensure a broader range of studies was included. However, studies that lacked robust experimental designs or did not report their results transparently were excluded from the final analysis. Data extraction from the included articles was carried out by two experienced researchers, L.F.L. and S.M.B., who have conducted systematic reviews and meta-analyses. Data extraction was performed using a standardized, systematic review method to thoroughly capture essential information on experimental models, details of Ligustilide treatment (including concentration and duration), outcomes, and limitations. We applied the PICO framework—Population, Intervention, Comparison, and Outcome—and analyzed constraints for the data extraction process. Additionally, we evaluated the design, sample size, and clarity of result reporting in the included studies to assess their quality, based on the experimental design and adherence to the PRISMA guidelines for scientific rigor [53]. A qualitative data synthesis was conducted to summarize the effects of Ligustilide on cancer cell lines and animal models. The goals were to identify expected outcomes and discuss limitations. The findings were organized into tables highlighting the anticancer effects of Ligustilide, its mechanisms of action, and its potential as a therapeutic agent. This review also aims to identify gaps in current research, suggest future directions for translational research initiatives, and promote the potential clinical applications of Ligustilide in cancer prevention and intervention.

### 4.2. Literature Search Report: Results of Literature Search Following PRISMA Guidelines

During the identification phase, 273 records were identified through database (n = 254) and register (n = 19) searches. Before screening, 180 records were removed—including 124 duplicate records, 54 records marked as ineligible by automation tools, and two records removed for other reasons. This left 93 records for screening. After the screening process, 64 records were excluded, leaving 29 reports to be retrieved; all were successfully retrieved (n = 0 reports not retrieved). These 29 reports were then assessed for eligibility. Of these, 10 reports were excluded for specific reasons: 4 were not in English, four did not involve Ligustilide, and 2 were non-experimental papers. Ultimately, 19 studies met the inclusion criteria and were included in the final review. Figure 3 depicts the PRISMA Flow Diagram for the above search [53]. Although articles were intended to be limited to English-language publications, the language of several records was not clearly identifiable during title/abstract screening. These records were excluded during the full-text assessment phase once it became evident that the full text was not available in English.

### 4.3. Preclinical Anticancer Studies of Ligustilide and (Z)-Ligustilide: Mechanisms, Efficacy, and Potential Clinical Implications

This section presents a comparative analysis of studies involving Ligustilide, categorized into two groups based on the specification of isomeric forms. The first table (Table 1) includes studies that did not specify the isomeric form of Ligustilide. In contrast, the second table (Table 2) focuses on studies identifying the Z-isomer of Ligustilide (Z-Ligustilide). To facilitate a clearer understanding of how Ligustilide and its Z-isomer may act across different biological contexts, the following subsections are organized by cancer type. This approach identifies potential variations in efficacy, mechanisms of action, or therapeutic relevance specific to certain tumor types. By comparing results within each cancer category, it becomes easier to observe patterns, assess consistency across studies, and highlight areas where isomer specification may influence experimental outcomes.

#### 4.3.1. Gastric Cancer

Gastric cancer remains one of the most common and lethal malignancies worldwide, often diagnosed at advanced stages [69]. It arises primarily from the stomach’s epithelial lining, with the most common histological subtype being adenocarcinoma [70]. The pathogenesis of gastric cancer involves genetic mutations, alterations in cellular signaling pathways (such as Wnt/β-catenin, PI3K/Akt, and MAPK), and the influence of environmental factors, such as *Helicobacter pylori* infection [71]. Tumor progression is often marked by EMT, resistance to apoptosis, and enhanced angiogenesis [72].

In the context of Ligustilide’s action on gastric cancer, in vitro studies on MKN74 and AGS cell lines, along with in vivo experiments in MKN74 xenograft-bearing nude mice, showed significant anticancer effects. Ligustilide treatment led to increased apoptosis, cell cycle arrest, and inhibited cell growth. These effects were associated with elevated caspase-3 activity, PARP cleavage, and upregulation of Bax, while Bcl-2 expression was downregulated, suggesting mitochondrial apoptosis. Furthermore, Ligustilide induced autophagy, as evidenced by increased LC3-II/LC3-I ratios and ATG5 expression, while reducing p62 levels. Additionally, it triggered endoplasmic reticulum stress pathways, including GRP78 activation, CHOP, and PERK phosphorylation, indicating a dual effect on apoptosis and autophagy. In vivo, Ligustilide administration reduced tumor volume and weight, and enhanced cleaved caspase-3 and p-PERK signaling, corroborating the results observed in cell culture models [15].

#### 4.3.2. Bile Duct Cancer

Bile duct cancer, or cholangiocarcinoma, is a rare but highly aggressive malignancy originating from the epithelial cells of the bile ducts [73,74]. The majority of cholangiocarcinomas present with poor prognosis due to late-stage diagnosis, with risk factors including chronic liver diseases, primary sclerosing cholangitis, and liver fluke infections [75]. The pathophysiology involves mutations in key genes such as KRAS, p53, and IDH1/2, as well as dysregulation of signaling pathways, including Wnt/β-catenin, Notch, and Hedgehog [76].

In vitro, Ligustilide was tested in two bile duct cancer cell lines, HuccT1 and RBE, and showed significant anticancer activity. The IC_50_ values—representing the concentration required to inhibit 50% of cell proliferation—were 5.08 µg/mL for HuccT1 cells and 5.77 µg/mL for RBE cells after 48 h of treatment. The compound demonstrated the ability to inhibit key processes associated with cancer progression, such as cell proliferation, migration, and invasion. Further investigation revealed that Ligustilide upregulated E-cadherin expression. This protein enhances cell adhesion while simultaneously downregulating N-cadherin, which is linked to EMT and cancer metastasis. Additionally, Ligustilide inhibited the PI3K/Akt signaling pathway, a critical pathway involved in cell survival and growth. In vivo, Ligustilide was administered to mice by intraperitoneal injection at 5 mg/kg for 18 days. The results indicated a significant reduction in tumor volume, suggesting that Ligustilide effectively suppresses tumor growth. Moreover, Ki67 expression, a marker of cell proliferation, was significantly downregulated in the tumors, further supporting the compound’s ability to inhibit cancer cell proliferation [14].

#### 4.3.3. Breast Cancer

Breast cancer is the most frequently diagnosed cancer in women and represents a significant cause of cancer-related deaths worldwide [77]. It is classified into several subtypes based on histological features and molecular biomarkers, with ER-positive, HER2-positive, and triple-negative breast cancer being the most common [78]. The pathophysiology of breast cancer involves mutations and aberrant activation of oncogenes (e.g., HER2, PIK3CA) and inactivation of tumor suppressors (e.g., BRCA1/2, p53), which lead to dysregulated cell proliferation, survival, and metastasis [79,80,81,82].

In the study by Alshehri et al. [54], Ligustilide showed promising anticancer effects in an Ehrlich solid carcinoma model in Sprague-Dawley rats. The results demonstrated reduced tumor weight and volume, accompanied by decreased cell proliferation. Mechanistically, Ligustilide suppressed Ki67 (a marker of proliferation) and mTOR signaling, suggesting an inhibition of cell growth. It also induced autophagy by activating beclin-1. These findings highlight Ligustilide’s potential as an effective agent for managing mammary tumors by modulating key signaling pathways involved in cell growth and survival.

(Z)-Ligustilide has shown promising effects in sensitizing breast cancer cells, including TAM-resistant cell lines, to treatment and modulating key autophagic and epigenetic pathways. Qi et al. [65] investigated the effects of (Z)-Ligustilide in TAM-resistant MCF-7^TR5^ and T47D^TR5^ cells, alongside their respective parental MCF-7 and T47D lines. The compound was found to sensitize TAM-resistant cells to apoptosis and to significantly affect autophagy, evidenced by increased LC3-II/LC3-I ratio and accumulation of RFP-LC3 puncta. (Z)-Ligustilide also restored the interaction between Nur77 and Ku80, proteins involved in apoptosis and autophagy, while increasing p62 protein levels and downregulating CTSD. These results suggest that (Z)-Ligustilide can overcome TAM resistance in breast cancer by modulating autophagic flux and apoptosis pathways.

In a separate study by Ma et al. [66], (Z)-Ligustilide was tested on triple-negative breast cancer cell lines, including MDA-MB-231, MDA-MB-453, and HS578t. The compound was shown to reactivate ERα protein expression in these ER-negative cell lines, restoring their sensitivity to TAM. Furthermore, (Z)-Ligustilide increased the Ace-H3 (lys9/14) within the ERα promoter region, while downregulating the expression of MTA1, IFI16, and HDAC, proteins involved in epigenetic repression. This epigenetic modulation suggests that (Z)-Ligustilide could be used to restore ERα activity in ER-negative breast cancer, potentially overcoming resistance to endocrine therapy.

Ligustilide and (Z)-Ligustilide both exhibit anticancer potential, but they work through different mechanisms. Ligustilide mainly inhibits tumor growth by reducing cell proliferation through the suppression of the Ki67 marker and mTOR signaling, while also promoting autophagy via beclin-1 activation. In contrast, (Z)-Ligustilide sensitizes TAM-resistant breast cancer cells to treatment. It modulates autophagy, restores apoptotic pathways, and reactivates ERα in triple-negative cells, overcoming TAM resistance. This occurs through epigenetic changes, including increased histone acetylation and downregulation of repressive proteins like MTA1 and HDAC. In summary, while Ligustilide targets tumor growth and survival pathways, (Z)-Ligustilide focuses on overcoming treatment resistance, particularly in hormone-resistant breast cancer. Together, they offer complementary therapeutic strategies.

#### 4.3.4. Bladder Cancer

Bladder cancer is a significant global health issue, with transitional cell carcinoma being the most common histological type [83]. Risk factors include smoking, chronic bladder infections, and exposure to industrial chemicals [84]. The pathophysiology of bladder cancer involves mutations in genes such as p53, RB1, and FGFR3, as well as alterations in critical signaling pathways, including MAPK, PI3K/Akt, and NF-κB [85,86]. This cancer type often shows resistance to chemotherapy [87], making novel therapeutic strategies like Ligustilide particularly relevant.

Yin et al. [55] studied Ligustilide’s anticancer effects on T24 and EJ-1 bladder cancer cells, observing significant inhibition of cell proliferation and induction of apoptosis. Ligustilide’s action was linked to upregulation of caspase-8, tBID and Bax and to downregulation of NF-κB1. In vivo, the compound also reduced tumor volume and weight. These findings suggest that Ligustilide promotes cancer cell death through mitochondrial regulation and NF-κB1-mediated apoptosis pathways, making it a potential candidate for bladder cancer therapy.

#### 4.3.5. Prostate Cancer

Prostate cancer is one of the most prevalent cancers in men, with risk factors including age, family history, and ethnicity [88]. It is primarily driven by AR-mediated signaling and mutations in genes such as PTEN, p53, and BRCA1/2 [89]. Despite advances in treatment, metastatic prostate cancer remains challenging, especially with the emergence of CRPC [90]. Therapeutic strategies targeting the tumor microenvironment, including CAF, have garnered increasing attention [91].

Ma et al. [56] explored the effects of Ligustilide on prostate CAF, specifically targeting their pro-angiogenesis and glycolytic activities. The treatment concentrations tested were 10, 20, and 40 µM, with exposure times of 1, 2, 4, 6, 12, 24, and 48 h. In vitro, the results showed significant suppression of CAF pro-angiogenic functions and a marked reduction in their glycolytic activity. In vivo, the study used a subcutaneous tumor model in C57BL/6 mice bearing RM-1 prostate cancer cells. The treatment was administered at a dose of 5 mg/kg via intraperitoneal injection for 18 days. The findings indicated a notable decrease in vascular density in the cancer tissue, which suggests an impairment in tumor angiogenesis. At the molecular level, the treatment induced phosphorylation of p38, ERK, and JNK, activating the TLR4-AP-1 signaling pathway. The study also observed downregulation of α-SMA and VEGFA expression, along with reduced levels of HIF-1 and several key glycolytic enzymes (HK1/2, GLUT1, PDK1, LDHA). On the other hand, p53 and Jab1 were upregulated. In summary, the treatment effectively disrupted the pro-angiogenic and glycolytic activities of CAF, leading to decreased vascular density and impaired tumor growth. This suggests that targeting CAF and their metabolic pathways could offer a promising therapeutic strategy for treating prostate cancer.

Ma et al. [57] investigated the effects of Ligustilide on prostate CAF and PC-3 cells, focusing on tumor growth inhibition via modulation of TLR4. The study used a range of concentrations from 0.01 to 0.32 mM, with an IC_50_ of 0.146 mM for CAF. The results showed a significant suppression of cell proliferation, increased apoptosis, and cell cycle arrest in vitro. In vivo, the treatment was administered to RM-1 tumor-bearing C57BL/6 mice and TLR4-deficient (TLR4^−/−^) mice at a dose of 5 mg/kg via intraperitoneal injection. In wild-type mice, the treatment led to a substantial reduction in tumor growth. Still, the effect was less pronounced in TLR4^−/−^ mice, highlighting the importance of TLR4 in mediating the treatment’s antitumor effects. Molecular analyses revealed that the treatment modulated the expression of cell cycle regulators, such as p21, cyclin B1, and cyclin D1, while promoting the activation of apoptotic pathways, including increased levels of Bax, cytochrome C, and cleaved caspase-3/-9. Additionally, there was a downregulation of the anti-apoptotic protein Bcl-2. The study also confirmed the modulation of TLR4 as a critical mechanism in this process. In summary, the study demonstrated that TLR4 modulation can inhibit prostate cancer tumor growth by promoting apoptosis and blocking proliferative signaling pathways. This highlights the potential of targeting TLR4 as a therapeutic strategy for prostate cancer treatment.

In their 2019 study, Ma et al. [58] focused on the immunomodulatory effects of Ligustilide on prostate CAF. The CAF were treated with concentrations of 15, 20, 30, and 45 µM for either 24 h or 4 days. The treatment resulted in a reversal of the immunosuppressive function of CAF, leading to the restoration of T-cell proliferation. This suggests that the therapy could potentially improve immune responses within the tumor microenvironment. Molecular investigations revealed that the treatment activated the NF-κB signaling pathway, which is known to play a role in immune regulation, and modulated TLR4 activity. In addition, there was a decrease in the expression of α-SMA, a marker of CAF activation, further indicating the treatment’s effect on CAF functionality. In summary, this study provides evidence that targeting the immunosuppressive function of CAF can enhance T-cell activity and potentially improve immunotherapy outcomes for prostate cancer. By modulating the NF-κB pathway and TLR4, the treatment could create a more favorable immune environment for antitumor responses.

(Z)-Ligustilide was tested in TRAMP C1 prostate cancer cells, showing a significant reduction in cell viability. At concentrations ranging from 6.25 to 100 µM over 1 to 5 days, (Z)-Ligustilide inhibited DNMT activity, particularly M.SssI, which plays a role in DNA methylation of the Nrf2 gene promoter. Additionally, (Z)-Ligustilide activated Nrf2 and its downstream antioxidant genes, such as HO-1 and NQO1. By modulating the epigenetic landscape and redox balance, (Z)-Ligustilide may enhance cellular stress responses, offering potential therapeutic benefits for prostate cancer [68].

(Z)-Ligustilide and Ligustilide differ primarily in their chemical isomerism, with (Z)-Ligustilide showing more specific therapeutic effects in prostate cancer treatment. Studies on (Z)-Ligustilide reveal its ability to inhibit DNMT activity, particularly affecting the Nrf2 gene promoter in prostate cancer cells. This activation leads to antioxidant pathways like HO-1 and NQO1, enhancing cellular stress responses in prostate cancer. In contrast, Ligustilide, while also possessing anti-inflammatory and anticancer properties, lacks the same detailed focus on epigenetic regulation and Nrf2 activation. As such, (Z)-Ligustilide appears to offer a more targeted approach, particularly in modulating epigenetic changes and redox balance in cancer cells, compared to the broader, less defined effects of Ligustilide.

#### 4.3.6. Hepatocellular Carcinoma

HCC is the most common type of liver cancer, often arising in the context of chronic liver disease, such as cirrhosis or hepatitis B and C infections [92]. It is associated with dysregulation of several oncogenic pathways, including the Wnt/β-catenin, PI3K/Akt/mTOR, and Hippo-YAP pathways [93,94,95]. Hepatic tumorigenesis is further driven by inflammatory cytokines like IL-6 and activation of STAT3 signaling [96].

In the study by Yang and Xing [59], Ligustilide demonstrated significant anticancer effects on HepG2 cells, a widely used model for HCC. It reduced cell viability and migration, while also inhibiting YAP activation and IL-6R/STAT3 signaling. This suggests that Ligustilide may effectively target the molecular drivers of HCC progression, particularly through modulation of the Hippo pathway and inflammatory signaling. Additionally, the compound’s ability to disrupt macrophage recruitment and skew toward the M2 phenotype indicates a potential immunomodulatory effect.

#### 4.3.7. Osteoblastoma

Osteoblastoma is a rare benign bone tumor characterized by excessive osteoblast growth [97]. Though commonly non-metastatic, it can cause significant local destruction and discomfort [98]. Zhang et al. [60] investigated the effects of Ligustilide on MG63 osteoblastoma cells. They found that it inhibited cell proliferation and induced apoptosis, with cell cycle arrest at the G2-M phase. The study highlighted Ligustilide’s ability to modulate TLR4 signaling and activate various apoptotic pathways, including p53, Bax, and caspase proteins. This suggests that Ligustilide may have therapeutic potential in treating osteoblastoma through immune modulation and apoptosis induction.

#### 4.3.8. Lung Cancer

Lung cancer, particularly NSCLC, remains one of the leading causes of cancer-related deaths globally [99]. It is often diagnosed at advanced stages due to the lack of early symptoms [100]. Mutations in EGFR, KRAS, and p53 are common, and the tumor microenvironment plays a crucial role in disease progression [101,102].

Jiang et al. [62] examined (Z)-Ligustilide’s effects on H1299 and A549 NSCLC cell lines, as well as A549 xenograft-bearing nude mice. (Z)-Ligustilide was found to inhibit cell proliferation, promote apoptosis, and suppress aerobic glycolysis, a hallmark of cancer metabolism. In vivo, (Z)-Ligustilide treatment significantly reduced tumor size and weight, likely due to modulation of the PTEN/Akt signaling pathway. These findings suggest that (Z)-Ligustilide could be a promising therapeutic agent for lung cancer, targeting both tumor growth and metabolic alterations.

Geng et al. [17] examined the effects of (Z)-Ligustilide on several NSCLC cell lines, including A549, A549/DDP (cisplatin-resistant), H460, and H460/DDP (cisplatin-resistant). The study evaluated the impact of (Z)-Ligustilide at concentrations of 15, 30, 60, 120, 180, and 240 µM over a 24 h exposure period. The results showed that (Z)-Ligustilide effectively inhibited cell viability in all tested cell lines. Additionally, the compound was found to reduce cisplatin resistance in the A549/DDP and H460/DDP cell lines, suggesting that it may enhance the sensitivity of these cells to cisplatin therapy through increased PLPP1 expression and suppression of the PIP3/Akt axis.

(Z)-Ligustilide shows significant anticancer effects. (Z)-Ligustilide targets cancer cell metabolism, inhibiting cell proliferation, promoting apoptosis, and suppressing glycolysis, with effects mediated through pathways like PTEN/Akt. It also modulates the tumor microenvironment, reducing tumor growth in NSCLC models [62]. (Z)-Ligustilide also enhances cisplatin sensitivity by lowering resistance in cisplatin-resistant NSCLC cell lines. Therefore, (Z)-Ligustilide effectively inhibits tumor progression, targets metabolic alterations and tumor growth, and shows promise in overcoming drug resistance, suggesting potential for combination therapies.

#### 4.3.9. Acute Myeloid Leukemia

AML is a hematologic malignancy marked by the uncontrolled proliferation of immature myeloid cells [103]. AML is characterized by genetic mutations that affect hematopoietic stem cell differentiation and survival [104]. Key mutations include FLT3, NPM1, CEBPA, and TP53 [105,106]. Dysregulation of signaling pathways such as the JAK-STAT, PI3K/Akt, and MAPK pathways further contributes to leukemogenesis [107]. The disease typically presents with a high burden of blasts in the bone marrow and peripheral blood, leading to neutropenia and increased susceptibility to infections and bleeding [108,109].

Chen et al. [16] studied the effects of (Z)-Ligustilide in AML cells, including HL-60, MV-4-11, and primary AML cells. In vitro, (Z)-Ligustilide showed IC_50_ values of 28.58 ± 2.53 µM for HL-60 cells and 25.37 ± 2.70 µM for MV-4-11 cells. It significantly reduced cell viability and induced ferroptosis, an iron-dependent form of cell death. The compound also disrupted iron metabolism, increasing ROS and lipid peroxidation while modulating key proteins involved in ferroptosis, such as IRP2, FTH1, ACSL4, and GPX4. In vivo, treatment with (Z)-Ligustilide decreased tumor growth, reduced peripheral white blood cell counts, improved inflammatory cell infiltration into the liver, and ameliorated hepatic function. These findings suggest that (Z)-Ligustilide induces ferroptosis and modulates the Nrf2/HO-1 pathway, presenting it as a potential therapeutic for AML.

Wang et al. [61] investigated the effects of (Z)-Ligustilide on AML cells, including HL-60, Kasumi-1, and MV-4-11, and its impact on cell survival and differentiation. In vitro, (Z)-Ligustilide showed IC_50_ values of 23.5 µM for HL-60, 36.1 µM for Kasumi-1, and 11.9 µM for MV-4-11 cells after 72 h of treatment. It significantly decreased cell viability, while promoting apoptosis (at higher concentrations) and cell differentiation (at lower concentrations). Notably, (Z)-Ligustilide restored the expression of Nur77 and NOR-1, key transcription factors involved in apoptosis, through histone acetylation. This led to increased p300 recruitment and decreased HDAC1, HDAC4/5/7, and MTA1 in the Nur77 promoter region. In vivo, the compound improved survival rates in NOD/SCID mice, reduced splenomegaly, and decreased white blood cell and lymphocyte counts. These findings suggest that (Z)-Ligustilide’s ability to restore Nur77 and NOR-1 expression may help overcome resistance mechanisms in AML.

#### 4.3.10. Oral Cancer

Oral cancer, including squamous cell carcinoma of the oral cavity, is often associated with risk factors such as tobacco use, alcohol consumption, and HPV infection [110]. The molecular pathogenesis involves genetic mutations in oncogenes like HRAS, p53, and CDKN2A, along with dysregulation of pathways such as the PI3K/Akt, MAPK, and Wnt/β-catenin [111,112]. The late-stage diagnosis and high rate of metastasis contribute to a poor prognosis, highlighting the need for more effective treatments [113].

In vitro studies on TW2.6, OML1, and SCC-25 cell lines demonstrated that (Z)-Ligustilide effectively inhibited cell migration and enhanced apoptosis. At concentrations ranging from 25 to 200 µM for 6 to 24 h, (Z)-Ligustilide also increased the radiosensitivity of cancer cells. Mechanistically, (Z)-Ligustilide activated endoplasmic reticulum-stress signaling and modulated HIF-1α and c-Myc expression, two key drivers of tumorigenesis and hypoxia responses. Additionally, it upregulated γ-H2AX, indicating DNA damage and repair modulation, which is crucial for therapeutic strategies targeting DNA repair in oral cancer [63].

#### 4.3.11. Ovarian Cancer

Ovarian cancer is one of the leading causes of cancer-related deaths in women [114]. Pathogenesis involves mutations in key tumor suppressors, such as BRCA1/2, p53, and PTEN, as well as activation of the PI3K/Akt/mTOR signaling pathway [115]. Ovarian cancer cells also frequently exhibit dysregulated DNA repair mechanisms, particularly through loss of homologous recombination repair due to BRCA1/2 mutations [116]. The disease is often diagnosed at advanced stages, making treatment challenging [117].

(Z)-Ligustilide’s effects on OVCAR-3 ovarian cancer cells were evaluated at concentrations of 50, 100, and 200 µM. The compound induced apoptosis and cell death through increased oxidative stress. Specifically, (Z)-Ligustilide triggered mitochondrial superoxide production and decreased mitochondrial polarization, leading to elevated ROS levels. These changes activated Nrf2, a critical regulator of antioxidant responses, along with its downstream target genes, including HO-1, NQO-1, and GCL. These results indicate that (Z)-Ligustilide may exploit oxidative stress pathways to induce cell death in ovarian cancer cells, suggesting potential therapeutic applications in the treatment of this malignancy [64].

#### 4.3.12. Glioblastoma

Glioblastoma is a highly aggressive and incurable brain tumor that often recurs despite current therapies. The pathogenesis involves alterations in key genetic pathways, including mutations in the tumor suppressor gene p53, PTEN, and EGFR. Activation of the PI3K/Akt and MAPK pathways, along with the upregulation of HIF signaling, plays a pivotal role in glioblastoma progression and resistance to treatment [118,119,120,121].

In vitro studies on T98G glioblastoma cells treated with (Z)-Ligustilide showed a significant reduction in cell mobility, single-cell migration, and wound-healing capacity. (Z)-Ligustilide also decreased the expression of Rho GTPases (RhoA, Rac1, Cdc42), which are crucial regulators of cell motility and invasion. These findings suggest that (Z)-Ligustilide may inhibit glioblastoma progression by disrupting signaling pathways involved in tumor cell migration and invasion, particularly by modulating Rho family GTPases [67,121]. A consolidated overview of the anticancer activities, experimental models, dosing parameters, and primary outcomes of Ligustilide and (Z)-Ligustilide is presented in Table 3.

## 5. Advanced Formulation Strategies for Ligustilide

Although Ligustilide shows very poor oral bioavailability (~2.6%) and chemical instability, several advanced formulation strategies have, however, been explored to overcome these limitations [25,27]. For instance, nanoemulsion systems have been used; a study of (Z)-Ligustilide nanoemulsions demonstrated significantly higher C_pmax_ and AUC_0→24 h_ after oral administration than free Ligustilide, as well as a pronounced anti-inflammatory efficacy [35].

More recently, nanoparticles were developed: co-encapsulation of Ligustilide with temozolomide in Poly (d,l-lactic-co-glycolide)-monomethoxy poly (ethylene glycol) nanoparticles improved stability (storage up to one month) and enhanced pharmacokinetics in rats, while also improving cytotoxicity against glioma cells [36]. In addition, chemical modification approaches have shown promise. A Ligustilide derivative, LIGc, was synthesized to improve stability; in pharmacokinetic studies, LIGc exhibited significantly higher oral bioavailability (~83.97%) than native Ligustilide [27]. Finally, emerging nanocrystal/self-stabilizing emulsion technologies have been applied: a spray-dried self-stabilizing nanocrystal emulsion increased intestinal absorption of Ligustilide in an ex vivo rat intestinal model [122].

Taken together, these advanced formulation and modification strategies—including nanoemulsions, nanoparticles, chemical derivatives, and nanocrystal-based systems—offer concrete approaches to improve both the pharmacokinetic profile and chemical stability of Ligustilide and should be further considered in the development of therapeutically viable formulations.

## 6. Recommendations for Clinical Translation

Advancing Ligustilide and (Z)-Ligustilide toward clinical application will require a structured translational framework. Before Phase I evaluation, essential preclinical studies should include comprehensive pharmacokinetic and pharmacodynamic characterization in multiple species, as well as ICH-compliant toxicology assessments to establish safety margins. Mechanistic validation studies are equally important: future work should incorporate pathway-specific inhibitors, gene knockdown or rescue assays, and epigenetic profiling to confirm causal roles for apoptosis, autophagy, ferroptosis, and HDAC/DNMT regulation. In vivo efficacy must be demonstrated across xenograft, orthotopic, and patient-derived xenograft models, particularly in drug-resistant tumors such as TAM-resistant breast cancer and cisplatin-resistant lung cancer. Additionally, combination studies should quantify drug–drug interactions using synergy metrics (e.g., the Chou–Talalay combination index) and integrate in vitro and in vivo confirmation.

Several biomarkers may serve as indicators of therapeutic response or mechanisms of action. These include restoring ERα expression in endocrine-resistant breast cancer, reducing HDAC and DNMT activity, and modulating PI3K/Akt/mTOR signaling, particularly changes in p-Akt, p-mTOR, and PTEN levels. Other promising markers include activation of the Nrf2/HO-1 oxidative stress response, alterations in ROS and lipid peroxidation associated with ferroptosis, and classical apoptotic indicators such as the Bax/Bcl-2 ratio and caspase activation. Tumor microenvironment–related biomarkers may also help define the compounds’ immunomodulatory and stromal effects. Incorporating these biomarkers into preclinical and early clinical studies may enable patient selection and pharmacodynamic monitoring.

Building on mechanistic insights, several rational drug combinations warrant investigation. (Z)-Ligustilide may be particularly effective when combined with TAM in endocrine-resistant breast cancer or with cisplatin in platinum-resistant lung cancer, where re-sensitization can be observed. Ligustilide’s modulation of PI3K/Akt/mTOR signaling suggests potential synergy with pathway-specific inhibitors, while its effects on immune regulation support combination approaches with checkpoint inhibitors. Radiosensitizing effects reported in oral cancer models also suggest possible benefits when combined with radiotherapy. Systematic evaluation of these combinations, including toxicity profiling and synergy assessment, will be essential.

These mechanistic and preclinical insights should guide early-phase clinical trial designs. A Phase I first-in-human study could employ a 3 + 3 or accelerated titration design in patients with refractory solid tumors, with an emphasis on breast and prostate cancers, where the strongest preclinical data exist. Primary endpoints would include safety, maximum tolerated dose, and pharmacokinetics. In contrast, exploratory endpoints should incorporate responsive biomarkers such as ERα restoration, HDAC/DNMT modulation, Nrf2/HO-1 activity, and circulating tumor DNA methylation signatures. A subsequent Phase II trial could adopt a basket design focused on endocrine-resistant breast cancer, cisplatin-resistant lung cancer, and prostate cancer, evaluating objective response rates, progression-free survival, and biomarker-based response correlations. Optional arms could evaluate rational combinations with endocrine therapy, platinum chemotherapy, or immunotherapy.

Together, these recommendations provide a structured translational pathway for Ligustilide and (Z)-Ligustilide and address the essential steps required to bridge preclinical promise with early clinical testing.

## 7. Conclusions

Ligustilide and its geometric isomer, (Z)-Ligustilide, represent promising phytochemicals with notable anticancer potential across a spectrum of malignancies. Preclinical evidence demonstrates that both compounds exhibit potent antitumor activity through multiple, often complementary mechanisms, including apoptosis induction, autophagy modulation, inhibition of cell proliferation, and modulation of oncogenic signaling pathways.

Overall, the evidence indicates that Ligustilide exhibits broader anticancer activity across multiple solid tumors—particularly gastric, bile duct, bladder, liver, and osteoblastoma cancer—where it consistently inhibits proliferation, induces apoptosis, and modulates key signaling pathways. In contrast, (Z)-Ligustilide demonstrates more specialized effects, especially in cancers driven by treatment resistance or epigenetic dysregulation, including breast cancer (TAM-resistant), AML, oral cancer, ovarian cancer, cisplatin-resistant lung cancer, and glioblastoma. (Z)-Ligustilide is particularly effective in overcoming drug resistance. Together, these findings suggest that Ligustilide is more broadly effective across diverse cancer types, whereas (Z)-Ligustilide shows more vigorous targeted activity in resistant or highly aggressive cancer phenotypes.

Comparative insights reveal that Ligustilide generally acts as a broad-spectrum cytotoxic and signaling-modulating agent. It inhibits tumor cell growth and survival by targeting pathways such as PI3K/Akt/mTOR, NF-κB, and STAT3, while promoting mitochondrial-mediated apoptosis and autophagy. Ligustilide also influences the tumor microenvironment by suppressing angiogenesis, modulating CAF, and restoring immune function—particularly through modulation of TLR4 and NF-κB signaling.

In contrast, (Z)-Ligustilide demonstrates more target-specific and mechanistically refined effects. It has shown superior potential in overcoming therapeutic resistance, notably in TAM-resistant breast cancer and cisplatin-resistant lung cancer models. Mechanistically, (Z)-Ligustilide’s actions often involve epigenetic reprogramming, including inhibition of HDACs, restoration of ERα expression, and suppression of DNMT activity. These effects re-sensitize resistant cancer cells to treatment. Moreover, (Z)-Ligustilide’s induction of ferroptosis and oxidative stress-mediated apoptosis, along with activation of antioxidant response elements (Nrf2/HO-1 pathway), underscores its dual role in redox regulation and cell death.

Together, these findings suggest that while Ligustilide exerts multi-pathway cytostatic and cytotoxic effects, (Z)-Ligustilide displays precision anticancer properties that may be harnessed for targeting drug-resistant or epigenetically deregulated tumors. Their complementary actions—one focusing on broad tumor suppression and the other on overcoming molecular resistance—highlight their potential for combination or sequential therapeutic strategies. Figure 4 illustrates the general overview of the anticancer mechanisms of Ligustilide and (Z)-Ligustilide, with Ligustilide primarily acting through mitochondrial stress and apoptosis, and (Z)-Ligustilide influencing epigenetic pathways and ferroptosis. Both compounds share outcomes such as apoptosis, growth inhibition, and tumor suppression, highlighting their complementary potential in cancer therapy.

Although preclinical evidence strongly supports the anticancer properties of Ligustilide and (Z)-Ligustilide, several key areas require further investigation to enable clinical translation. First, mechanistic clarification is essential. Employing transcriptomic, proteomic, and metabolomic approaches will help identify unique molecular targets and biomarkers predictive of therapeutic response.

Second, the pharmacokinetic limitations of these compounds must be addressed. Both Ligustilide and (Z)-Ligustilide are chemically unstable and exhibit poor bioavailability in vivo. Innovative formulation strategies—such as nanoparticle delivery, liposomal encapsulation, or chemical derivatization—could enhance their stability, solubility, and systemic retention.

Third, their potential as combination agents should be explored. (Z)-Ligustilide’s demonstrated ability to reverse TAM and cisplatin resistance suggests strong potential in multi-drug regimens with conventional chemotherapeutics, targeted therapies, or immune checkpoint inhibitors. Systematic studies on synergistic interactions could pave the way for optimized combination protocols.

It should be noted, however, that although several studies report that (Z)-Ligustilide can re-sensitize TAM-resistant breast cancer and cisplatin-resistant lung cancer cells, the available evidence is based mainly on in vitro models and molecular markers of restored sensitivity. Most studies did not include formal quantification of synergy or systematic in vitro/in vivo validation. Therefore, the current findings should be interpreted as preliminary indications of resistance reversal rather than definitive proof. To substantiate the potential of (Z)-Ligustilide to overcome therapeutic resistance, future research should incorporate rigorous validation methods, including combined in vitro/in vivo studies and synergy index calculations.

Fourth, greater attention should be given to their effects on the tumor microenvironment and immune modulation. Ligustilide’s role in regulating CAF activity, angiogenesis, and T-cell restoration warrants deeper investigation in immunocompetent and patient-derived xenograft models. Such studies could clarify how these compounds influence immune homeostasis and tumor–stroma crosstalk.

Finally, translational studies are critical for advancing these findings toward clinical application. Comprehensive toxicological assessments, pharmacodynamic evaluations, and formulation of standardized dosing protocols are prerequisites for early-phase clinical trials. Future clinical studies should prioritize cancer types with the strongest preclinical responses, including breast and prostate cancers, for which both compounds have demonstrated significant efficacy and mechanistic depth.

To strengthen clinical translation, future research should prioritize cancer types for which the strongest mechanistic and preclinical evidence currently exists—specifically breast cancer and prostate cancer. These malignancies show consistent responsiveness to both Ligustilide and (Z)-Ligustilide across apoptosis-, autophagy-, and epigenetic-related pathways, making them high-yield candidates for early-phase clinical evaluation. In addition, structured collaborative research models will accelerate progress. Multi-institutional consortia integrating molecular oncology, pharmacology, and medicinal chemistry could streamline standardized comparisons of the compound. Partnerships with industry may support formulation engineering to overcome instability and bioavailability limitations, while multi-center, biomarker-guided preclinical and early clinical studies would enable validation of predictive markers such as ERα restoration, HDAC inhibition, or Nrf2 targeting. Incorporating these concrete priorities and collaborative frameworks will facilitate the efficient advancement of Ligustilide and (Z)-Ligustilide toward clinical testing.

Across the included studies, Ligustilide and (Z)-Ligustilide were consistently shown to modulate several major oncogenic pathways, although not every study validated all mechanisms to the same extent. Suppression of the PI3K/Akt axis was observed in cholangiocarcinoma and lung cancer models, accompanied by increased PTEN expression and reduced Akt phosphorylation [14,62]. Modulation of the MAPK family kinases, particularly ERK, JNK, and p38, has been repeatedly reported in prostate cancer-associated fibroblasts and osteoblastoma cells through TLR4 and AP-1-mediated mechanisms [56,57,58,60]. Several studies also documented p53-mediated apoptosis, including increased Bax, cytochrome-c release, and caspase activation across prostate, bladder, and osteoblastoma models [55,57,60].

The Nrf2/HO-1 oxidative-stress pathway was among the most consistently modulated mechanisms, with Nrf2 accumulation and upregulation of HO-1, NQO1, and related enzymes demonstrated in AML, ovarian, and prostate cancer cells [16,64,68]. Regulation of autophagy was also frequently observed, including increased LC3-II/LC3-I ratios, altered p62 levels, and Beclin-1 activation in gastric, breast, and mammary tumor models [15,54,65]. Certain mechanisms appeared cancer-type specific, such as ferroptosis induction in AML [16] and endoplasmic reticulum–stress-driven radiosensitization in oral cancer [63].

Overall, these findings suggest that Ligustilide affects multiple convergent signaling pathways; however, some studies reported on correlative molecular changes without pathway-specific inhibitors, gene knockdown/overexpression, or rescue experiments. Thus, while the mechanistic patterns are compelling, further targeted validation is needed to confirm the causal contributions of each pathway.

In conclusion, Ligustilide and (Z)-Ligustilide exemplify how subtle structural variations within a natural compound can result in distinct and complementary anticancer mechanisms. Ligustilide primarily acts by broadly regulating tumor growth, survival, and the tumor microenvironment, whereas (Z)-Ligustilide exhibits greater specificity in reversing drug resistance and modulating epigenetic and oxidative stress pathways. The integration of these mechanistic insights with advancements in formulation science and systems pharmacology could accelerate the transition of Ligustilide from preclinical promise to clinical utility. Continued interdisciplinary collaboration—bridging pharmacology, molecular oncology, and medicinal chemistry—will be vital to fully realize the therapeutic potential of Ligustilide and its (Z)-isomer in combating cancer development and progression.

## Figures and Tables

**Figure 1 pharmaceuticals-19-00036-f001:**
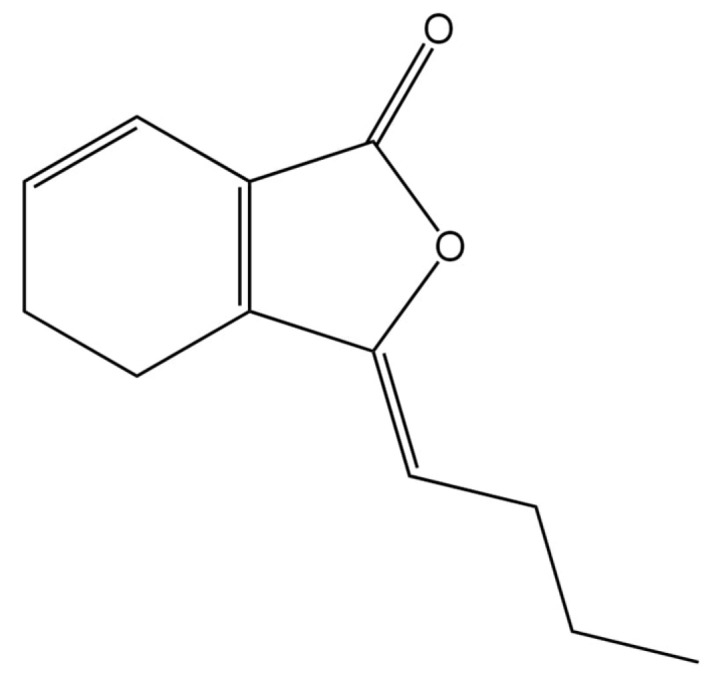
Ligustilide’s Molecular Structure.

**Figure 2 pharmaceuticals-19-00036-f002:**
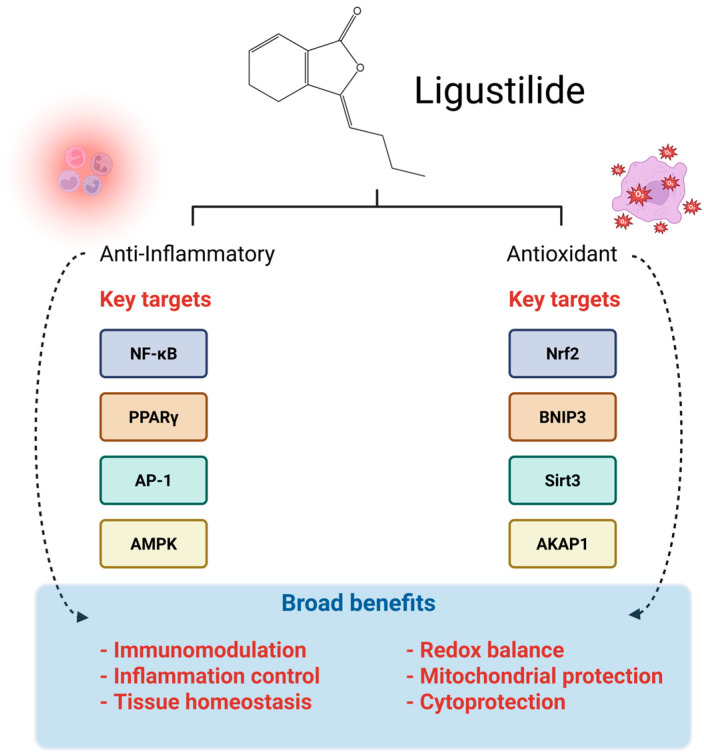
Dual Pharmacological Actions of Ligustilide. Ligustilide exerts both anti-inflammatory and antioxidant effects by modulating multiple cellular pathways. On the anti-inflammatory side, it modulates key inflammatory mediators, including NF-κB, PPARγ, AP-1, and the AMPK/GSK-3β axis, contributing to immunomodulation and inflammation resolution. On the antioxidant side, Ligustilide enhances cellular antioxidant defenses via Nrf2 activation, mitophagy induction (BNIP3–LC3), Sirt3-mediated mitochondrial homeostasis, and PKA/AKAP1 signaling. These dual effects support its therapeutic potential across diverse systems, including the nervous, cardiovascular, immune, and integumentary systems. They promote redox balance, tissue protection, and homeostasis in conditions associated with oxidative stress and inflammation. Created in BioRender (https://biorender.com/9aifgay), last accessed on 23 October 2025.

**Figure 3 pharmaceuticals-19-00036-f003:**
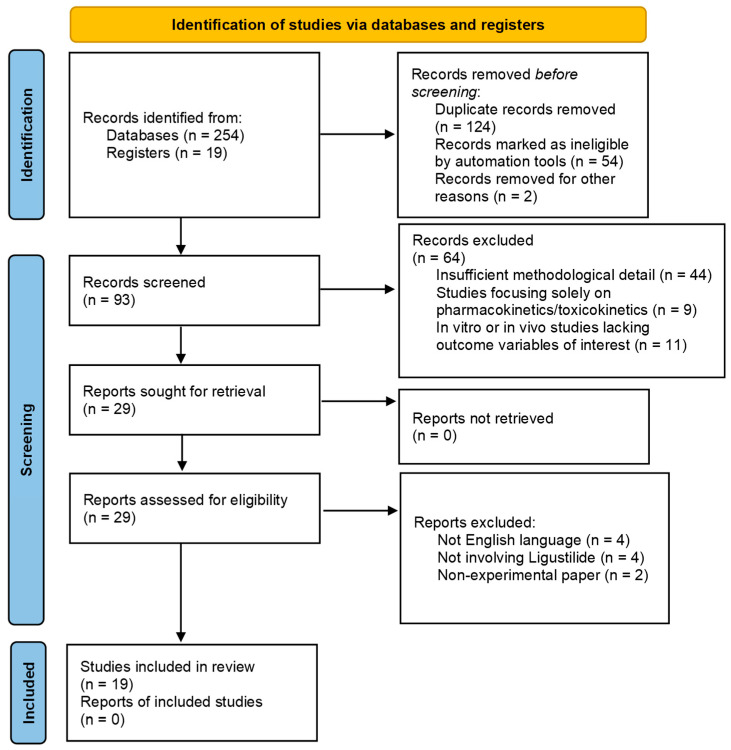
Report of the Included Studies following the PRISMA Guidelines [53].

**Figure 4 pharmaceuticals-19-00036-f004:**
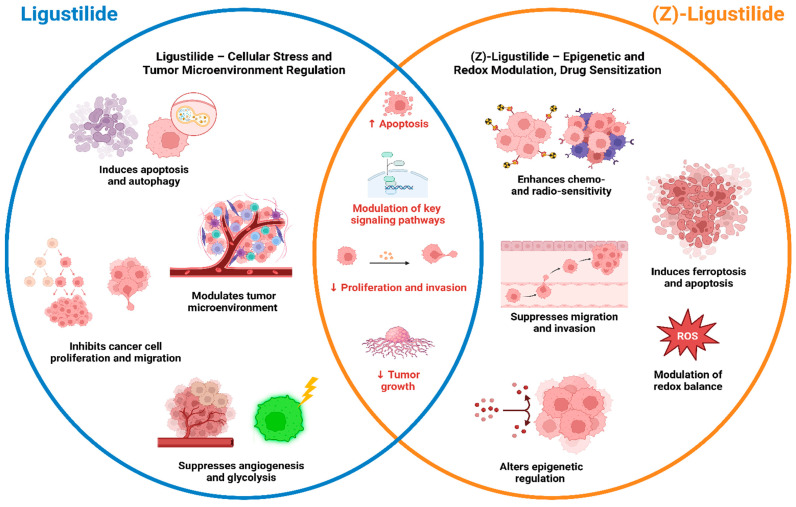
General overview of the anticancer mechanisms of Ligustilide and (Z)-Ligustilide. Ligustilide predominantly acts through mitochondrial stress, apoptosis, and microenvironment modulation, while (Z)-Ligustilide primarily influences epigenetic pathways, enhancing drug sensitivity and ferroptosis. Both compounds converge on shared outcomes—apoptosis, growth inhibition, and tumor suppression—highlighting their complementary potential in cancer therapy. Created in BioRender (https://biorender.com/cnjeeq0), last accessed on 23 October 2025. ↑: Increase; ↓: Decrease.

**Table 1 pharmaceuticals-19-00036-t001:** Anticancer effects and underlying mechanisms of action of Ligustilide against different types of cancer across various cellular and animal models.

Cell Line (s)/Animal Model (s)	IC_50/_EC_50/_Concentration and Duration	Effects Demonstrated	Mechanisms of Action	Reference
**Gastric cancer**				
In vitro: MKN74, AGS. In vivo: tumor xenograft (MKN74 cells)-bearing nude mice.	In vitro: 5, 10, 20, 40, 60, 80, 100, 200 and 300 µM, 24 and 48 h. In vivo: 5 mg/kg (intraperitoneal injection) for 20 days.	In vitro: ↑ apoptosis and cell cycle arrest, **┴** cell growth. In vivo: ↓ tumor volume and weight.	In vitro: ↑ caspase-3 activity and PARP cleavage, ↑ Bax, ↓ mitochondrial Bcl-2, ↑ autophagy (↑ LC3-II/LC3-I ratio, ↓ p62 protein levels, ↑ ATG5 expression), upregulation of endoplasmic reticulum stress markers (GRP78, CHOP), phosphorylation of PERK. In vivo: ↑ cleaved caspase-3 and p-PERK signals, ↑ autophagy (LC3).	Liu et al. (2025) [15]
**Bile duct cancer**				
In vitro: HuccT1 and RBE. In vivo: NOG mice with cholangiocarcinoma.	In vitro: 5.08 µg/mL, 48 h (IC_50_—HuccT1 cells); 5.77 µg/mL, 48 h (IC_50_—RBE cells). In vivo: 5 mg/kg (intraperitoneal injection) for 18 days.	In vitro: **┴** cell proliferation, migration and invasion. In vivo: ↓ tumor volume.	In vitro: ↑ E-cadherin expression, ↓ N-cadherin expression, ↑ NDRG1, ┴ PI3K/Akt signaling pathway. In vivo: downregulation of Ki67 expression.	Wu et al. (2025) [14]
**Mammary tumor**				
In vivo: Sprague-Dawley rats with Ehrlich solid carcinoma.	In vivo: 20 mg/kg (oral gavage) for 3 weeks.	In vivo: ↓ tumor weight and volume, **┴** cell proliferation, **┴** apoptosis.	In vivo: ┴ Ki67 and mTOR, ↓ AMPK expression, ↑ Bcl-2, ↑ autophagy (beclin 1 activation).	Alshehri et al. (2023) [54]
**Bladder cancer**				
In vitro: T24 and EJ-1. In vivo: xenograft tumor (T24 and EJ-1 cells)-bearing nude mice.	In vitro: 209.8 µM, 24 h (IC_50_—T24 cells); 215.2 µM, 48 h (IC_50_—T24 cells); 240.4 µM, 24 h (IC_50_—EJ-1 cells); 230.3 µM, 48 h (IC_50_—EJ-1 cells). In vivo: 10 mg/kg (intraperitoneal injection) every 3 days for 27 or 30 days.	In vitro: **┴** cell proliferation and the cell cycle at the sub-G1 phase, ↑ apoptosis. In vivo: ↓ tumor volume and weight.	In vitro: upregulation of the expression of caspase-8, tBID and Bax proteins, downregulation of the expression of NF-κB1 (p50) protein. In vivo: promotes cancer cell death through mitochondrial regulation and NF-κB1-mediated pathways.	Yin et al. (2023) [55]
**Prostate cancer**				
In vitro: prostate CAF. In vivo: subcutaneous tumor (RM-1 cells)-bearing C57BL/6 mice.	In vitro: 10, 20 and 40 µM, 1, 2, 4, 6, 12, 24 and 48 h. In vivo: 5 mg/kg (intraperitoneal injection) daily for 18 days.	In vitro: **┴** pro-angiogenesis effect of CAF, **┴** glycolysis in CAF. In vivo: ↓ vascular density in cancer tissue.	In vitro: phosphorylation of p38, ERK and JNK, activation of the TLR4-AP-1 signaling pathway, ↓ expression levels of α-SMA and VEGFA, **┴** HIF-1, ↓ HK1/2, GLUT1, PDK1, LDHA, upregulation of p53 and Jab1. In vivo: ↓ expression levels of α-SMA, CD31, VEGFA and HGF.	Ma et al. (2022) [56]
In vitro: prostate CAF and PC-3. In vivo: tumor (RM-1 cells)-bearing C57BL/6 mice and TLR4^−/−^ mice.	In vitro: 0.146 mM (IC_50_, CAF); 0.01, 0.02, 0.04, 0.08, 0.16, 0.24 and 0.32 mM, 0.5, 1, 2, 4, 8, and 24 h. In vivo: 5 mg/kg (intraperitoneal injection).	In vitro: ┴ cell proliferation, ↑ apoptosis and cell cycle arrest. In vivo: ┴ tumor growth.	In vitro: modulation of p21, cyclin B1 and cyclin D1, ↑ phosphorylation of p53, ↑ Bax, cytochrome C and cleaved caspase-3/-9, downregulation of Bcl-2, modulation of TLR4. In vivo: modulation of TLR4.	Ma et al. (2020) [57]
In vitro: prostate CAF.	In vitro: 15, 20, 30 and 45 µM, 24 h and 4 days.	In vitro: reversion of the immunosuppressive function of CAF and restoration of T-cell proliferation.	In vitro: activation of the NF-κB pathway, modulation of TLR4, ↓ α-SMA.	Ma et al. (2019) [58]
**Hepatocellular carcinoma**				
In vitro: HepG2.	In vitro: 2.5, 5, 10, 20, 50, 100 and 200 µM, 24 h.	In vitro: ┴ cell viability and migration, ↓ cancer cell malignancy.	In vitro: ┴ YAP activation, ↓ IL-6 release, ┴ IL-6R/STAT3 signaling activation, ┴ cancer cells’ ability to recruit and skew macrophages toward M2 phenotype.	Yang & Xing (2021) [59]
**Osteoblastoma**				
In vitro: MG63.	In vitro: 0.294 mM (IC_50_).	In vitro: ┴ cell proliferation, ↑ apoptosis, arrested the cell cycle in G2-M phase.	In vitro: modulation of TLR4, upregulation of p-p53, p21, cyclin D1, p-Tak1, p-ERK and Bax, downregulation of p53, cyclin B1, Tak1 and ERK, activation of Caspase family.	Zhang et al. (2022) [60]

**Note:** Various symbols (↑, ↓, and ┴) indicate an increase, a decrease, and an inhibition in the obtained variables, respectively.

**Table 2 pharmaceuticals-19-00036-t002:** Anticancer effects and underlying mechanisms of action of (Z)-Ligustilide against different types of cancer across various cellular and animal models.

Cell Line (s)/Animal Model (s)	IC_50/_EC_50/_Concentration and Duration	Effects Demonstrated	Mechanisms of Action	Reference
**Acute myeloid leukemia**				
In vitro: HL-60, MV-4-11 and primary AML cells. In vivo: HL-60 cells injected in BALB/c-nu nude mice.	In vitro: 28.58 ± 2.53 µM (IC_50_—HL-60 cells); 25.37 ± 2.70 µM (IC_50_—MV-4-11 cells); 6.25, 12.5, 25, 30, 50, 70 and 100 µM, 6, 24, 48 and 72 h. In vivo: 40 mg/kg/2 days (intraperitoneal injection) for 12 days.	In vitro: ┴ cell viability, promotion of iron metabolism disorder, ↑ cell death and ferroptosis. In vivo: ↓ tumor growth, ↓ white blood cell counts in the peripheral blood of mice, improved inflammatory cell infiltration into the liver and hepatic sinusoidal contraction.	In vitro: modulation of Nrf2/HO-1 pathway, ↑ ROS and lipid peroxidation, ↓ IRP2 protein and TRF1 expression, ↑FTH1 expression, ↑ ACSL4 and PTGS2 protein levels, ↓ GPX4 levels. In vivo: ↑ Nrf2 and HO-1 proteins.	Chen et al. (2024) [16]
In vitro: HL-60, Kasumi-1 and MV-4-11. In vivo: HL-60 cells injected in NOD/SCID mice.	In vitro: 23.5 µM, 72 h (IC_50_—HL-60 cells); 36.1 µM, 72 h (IC_50_—Kasumi-1 cells); 11.9 µM, 72 h (IC_50_—MV-4-11 cells). In vivo: 80 mg/kg (intraperitoneal injection) once every other day for 2 weeks.	In vitro: ┴ cell viability, ↑ apoptosis (at higher concentrations of Z-Ligustilide) and cell differentiation (at lower concentrations of Z-Ligustilide). In vivo: ↑ mice survival rate, ↓ splenomegaly, ↓ white blood cell and lymphocyte counts in mice.	In vitro: restoration of Nur77 and NOR-1 expression through histone acetylation, ↑ recruitment of p300, ↓ recruitment of HDAC1, HDAC4/5/7 and MTA1 in the Nur77 promoter region, ↓ HDAC1 and HDAC3 in the NOR-1 promoter region. In vivo: restoration of Nur77 and NOR-1.	Wang et al. (2021) [61]
**Lung cancer**				
In vitro: A549, A549/DDP (cisplatin-resistant), H460 and H460/DDP (cisplatin-resistant).	In vitro: 15, 30, 60, 120, 180 and 240 µM, 24 h.	In vitro: ┴ cell viability, ↓ cisplatin resistance of A549/DDP and H460/DDP.	In vitro: (Z)-Ligustilide plus cisplatin induced ↑ PLPP1 expression and ┴ PIP3/Akt axis.	Geng et al. (2023) [17]
In vitro: H1299 and A549. In vivo: BALB/c nude mice with orthotopic tumor (A549 cells).	In vitro: 15, 30, 60, 120 and 180 µM, 12, 24 and 48 h. In vivo: 5 mg/kg (intraperitoneal injection).	In vitro: ┴ cell proliferation, ↑ apoptosis, ↓ aerobic glycolysis of the cells. In vivo: ↓ tumor size, volume and weight.	In vitro: upregulation of PTEN, ┴ phosphorylation of Akt, ↑ caspase-3/-7 activity, downregulation of GLUT1, HK1/2, LDHA and PDK1. In vivo: ↓ percentage of Ki-67-positive cells in tumor tissues, modulation of PTEN/Akt signaling pathway.	Jiang et al. (2021) [62]
**Oral cancer**				
In vitro: TW2.6, OML1 and SCC-25.	In vitro: 25, 50, 100 and 200 µM, 6, 16 and 24 h.	In vitro: ↑ apoptosis, ┴ cell migration, ↑ cancer’s radiosensitivity.	In vitro: activation of endoplasmic reticulum-stress signaling, modulation of HIF-1α, ↑ c-Myc protein levels and cleaved caspase-3, ↑ γ-H2AX expression.	Hsu et al. (2022) [63]
**Ovarian cancer**				
In vitro: OVCAR-3.	In vitro: 50, 100 and 200 µM.	In vitro: ↑ apoptosis and total cell death, ↑ oxidative stress.	In vitro: ↑ mitochondrial superoxide formation, ↓ mitochondrial polarization, ↑ ROS, ↑ nuclear level of Nrf2 and its downstream target genes (HO-1, NQO-1, UGT1A1, GCL).	Lang et al. (2018) [64]
**Breast cancer**				
In vitro: MCF-7, MCF-7^TR5^ (TAM-resistant), T47D and T47D^TR5^ (TAM-resistant).	In vitro: 25, 50, 100 and 200 µM, 3, 6, 12, 24 and 48 h.	In vitro: sensitizes TAM-resistant cells to apoptosis, ┴ autophagy and autophagosome-lysosome fusion in MCF-7^TR5^.	In vitro: restoration of the interaction between Nur77 and Ku80, ↑ LC3-II/LC3-I ratio and accumulation of RFP-LC3 puncta, ↑ p62 protein level, downregulation of CTSD.	Qi et al. (2017) [65]
In vitro: MDA-MB-231, MDA-MB-453 and HS578t.	In vitro: 10, 25, 50 µM, 12, 24, 36, 48 and 72 h. 133.6 µM (IC_50_—MDA-MB-231 cells).	In vitro: reactivation of ERα protein expression and restoration of cells’ sensitivity to TAM.	In vitro: ↑ Ace-H3 (lys9/14) level in the ERα promoter region, ↓ MTA1, IFI16 and HDAC expression.	Ma et al. (2017) [66]
**Glioblastoma**				
In vitro: T98G.	In vitro: 2.5, 5, 10 and 25 µM, 14 and 20 h.	In vitro: ↓ cell mobility, single cell migration and wound-like gap closure capacity.	In vitro: ↓ expression levels of the Rho GTPases (RhoA, Rac1, Cdc42).	Yin et al. (2013) [67]
**Prostate cancer**				
In vitro: TRAMP C1.	In vitro: 6.25, 12.5, 20, 25, 40, 50, 60, 80 and 100 µM, 1, 3 and 5 days. IC_50_: 1055 µM (specific for inhibition of M.SssI activity).	In vitro: ↓ cell viability.	In vitro: ↑ Nrf2 and Nrf2-mediated enzymes (HO-1, NQO1, UGT1A1), ┴ DNMT activity of M.SssI, ↓ methylated CpG ratio in the Nrf2 gene promoter region.	Su et al. (2013) [68]

**Note:** Various symbols (↑, ↓, and ┴) indicate an increase, a decrease, and an inhibition in the obtained variables, respectively.

**Table 3 pharmaceuticals-19-00036-t003:** Consolidated overview of the experimental models, dosing parameters, and primary outcomes of Ligustilide and (Z)-Ligustilide.

Cancer Type	Model Type	Model Used	Dose/Exposure	Primary Outcome	Reference
Gastric cancer	In vitro/In vivo	MKN74, AGS; xenograft mice	5–300 µM; 5 mg/kg i.p.	Apoptosis induction; ↓ tumor volume	Liu et al. (2025) [15]
Bile duct cancer	In vitro/In vivo	HuccT1, RBE; NOG mice	~5 µg/mL; 5 mg/kg i.p.	┴ proliferation and migration; ↓ tumor volume	Wu et al. (2025) [14]
Mammary tumor	In vivo	Sprague-Dawley rats	20 mg/kg oral	↓ tumor weight/volume	Alshehri et al. (2023) [54]
Bladder cancer	In vitro/In vivo	T24, EJ-1; xenograft mice	~209–240 µM; 10 mg/kg i.p.	↑ apoptosis; ↓ tumor size	Yin et al. (2023) [55]
Prostate cancer	In vitro/In vivo	CAF, PC-3; C57BL/6 and TLR4^−/−^ mice	10–45 µM and 0.01–0.32 mM; 5 mg/kg i.p.	Anti-angiogenic; ↓ tumor growth; immunomodulation	Ma et al. (2022; 2020; 2019) [56,57,58]
Hepatocellular carcinoma	In vitro	HepG2	2.5–200 µM	┴ viability and migration	Yang & Xing (2021) [59]
Osteoblastoma	In vitro	MG63	~0.3 mM	↑ apoptosis	Zhang et al. (2022) [60]
Lung cancer	In vitro/In vivo	A549, H1299; nude mice with orthotopic tumor	15–180 µM; 5 mg/kg i.p.	┴ proliferation; ↓ tumor growth	Jiang et al. (2021) [62]
AML	In vitro/In vivo	HL-60, MV-4-11, primary AML cells, Kasumi-1; BALB/c mice, NOD/SCID mice	6.25–100 µM; 40–80 mg/kg	Ferroptosis induction; ↓ leukemic burden	Chen et al. (2024); Wang et al. (2021) [16,61]
Lung cancer (cisplatin-resistant)	In vitro	A549/DDP, H460/DDP	15–240 µM	↓ cisplatin resistance	Geng et al. (2023) [17]
Oral cancer	In vitro	TW2.6, OML1, SCC-25	25–200 µM	↑ apoptosis; ↑ radiosensitivity	Hsu et al. (2022) [63]
Ovarian cancer	In vitro	OVCAR-3	50–200 µM	↑ apoptosis; ↑ oxidative stress	Lang et al. (2018) [64]
Breast cancer	In vitro	MCF-7, T47D, and TAM-resistant lines; MDA-MB-231, MDA-MB-453, and HS578t	10–200 µM	Restores TAM sensitivity; ↑ ERα expression	Qi et al. (2017); Ma et al. (2017) [65,66]
Glioblastoma	In vitro	T98G	2.5–25 µM	↓ migration	Yin et al. (2013) [67]
Prostate cancer	In vitro	TRAMP C1	6.25–100 µM	↑ Nrf2 signaling; ┴ DNMT activity	Su et al. (2013) [68]

**Note:** Various symbols (↑, ↓, and ┴) indicate an increase, a decrease, and an inhibition in the obtained variables, respectively.

## Data Availability

No new data were created or analyzed in this study. Data sharing is not applicable to this article.

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
