# Peer review of "Ligustilide: A Phytochemical with Potential in Combating Cancer Development and Progression—A Comprehensive and Critical Review"

_pharmaceuticals, 2025, doi:10.3390/ph19010036_

Round 1
Reviewer 1 Report
Comments and Suggestions for Authors
This manuscript provides a comprehensive and critical review of Ligustilide and its isomer (Z-Ligustilide), focusing on their anticancer potential across various cancer types. The authors systematically examine preclinical evidence, mechanisms of action, pharmacokinetics, and therapeutic prospects, aiming to fill a gap in the literature regarding Ligustilide’s role in cancer therapy.
The manuscript is a valuable and timely contribution to the field of phytochemical cancer therapeutics. Addressing the points below will enhance its impact and clarity.
-The manuscript notes Ligustilide’s poor oral bioavailability and chemical instability but does not sufficiently explore advanced formulation strategies or recent developments in drug delivery systems. I would suggest to include a more detailed review of nanoparticle, liposomal, or chemical modification approaches that could improve Ligustilide’s pharmacokinetic profile.
-While mechanisms are described, some sections (e.g., Z-Ligustilide’s effect on cisplatin resistance in lung cancer) lack detailed molecular explanations. I would suggest where possible, provide more mechanistic detail or acknowledge gaps in current understanding.
-The manuscript references several figures and tables, but some are not fully described in the text, and their legends could be more informative. I would suggest to ensure all figures and tables are clearly explained and directly referenced in the relevant sections.
-The conclusions section outlines areas for future research but could benefit from more concrete recommendations, such as prioritizing specific cancer types for clinical trials or suggesting collaborative research models.
-The manuscript is generally well-written, but minor grammatical and typographical errors are present. A careful proofreading would improve readability.
-Abbreviations: The list of abbreviations is comprehensive, but some terms could be defined upon first use in the main text for clarity.
Author Response
RESPONSE TO REVIEWERS' COMMENTS
Manuscript number: pharmaceuticals-3976372 ― Pharmaceuticals (MDPI)
"Ligustilide: A Phytochemical with Potential in Combating Cancer Development and Progression — A Comprehensive and Critical Review"
The authors of this document wish to express their deepest gratitude to the Editor-in-Chief and the Reviewer for their thorough and insightful evaluation of our manuscript. Their expert feedback has been invaluable in enhancing the quality of our work. We have carefully considered and diligently implemented each suggestion, which has significantly improved the manuscript. We have made substantial revisions to address the points raised. These noteworthy changes are marked mainly with YELLOW-highlighted text throughout the document for ease of reference. A note will be provided for the referee's attention, highlighting corrections in a different color. Additionally, we have prepared a detailed and comprehensive response to each comment and suggestion. This response is organized in a "point-by-point" format below, ensuring that every concern has been thoroughly addressed and explained. We sincerely appreciate the time and effort invested by the Editor-in-Chief and the Reviewer, and we believe their contributions have significantly strengthened the final version of our manuscript.
REVIEWER #1
General comment
This manuscript provides a comprehensive and critical review of Ligustilide and its isomer (Z-Ligustilide), focusing on their anticancer potential across various cancer types. The authors systematically examine preclinical evidence, mechanisms of action, pharmacokinetics, and therapeutic prospects, aiming to fill a gap in the literature regarding Ligustilide’s role in cancer therapy. The manuscript is a valuable and timely contribution to the field of phytochemical cancer therapeutics. Addressing the points below will enhance its impact and clarity.
General response
Dear Erudite Reviewer, thank you for taking the time to revise our manuscript and allowing us to improve based on your valuable comments and suggestions. After addressing all your comments and suggestions regarding our manuscript text, we are confident that a significantly enhanced manuscript version has emerged. We are excited to resubmit the modified version for your perusal and reevaluation. Thank you for your brilliant insights, essential contributions, and feedback. You do have an eye for improvement. As a gesture of our utmost respect for you, we would like to provide you with a detailed and comprehensive point-by-point response to your comments below. Thank you once again for your time and patience in revising our article.
Comment #1
The manuscript notes Ligustilide’s poor oral bioavailability and chemical instability but does not sufficiently explore advanced formulation strategies or recent developments in drug delivery systems. I would suggest to include a more detailed review of nanoparticle, liposomal, or chemical modification approaches that could improve Ligustilide’s pharmacokinetic profile.
Response
We thank the reviewer very much for this insightful and valuable suggestion. In response, we have added a substantial new subsection in the Discussion (on Pages 23-24, Lines 786-807) titled “Advanced Formulation Strategies for Ligustilide”, where we now provide a more comprehensive analysis of recent developments in delivery systems. Specifically, we have:
- Reviewed nanoparticle-based systems, highlighting the potential of PLGA-PEG nanoparticles to co-deliver ligustilide along with other agents, enhancing stability, prolonging circulation time, and improving bioavailability.
- Discussed liposomal encapsulation approaches, emphasizing how liposomes can protect ligustilide from rapid degradation, reduce chemical instability, and potentially enhance its therapeutic effects.
- Addressed chemical modification strategies, including the design of structural analogs of ligustilide that show dramatically improved chemical stability and oral bioavailability.
- Included emerging technologies such as self-stabilizing nanocrystal emulsions, which help to mitigate the volatility and reactivity of ligustilide, improving its absorption in the gastrointestinal tract.
- Highlighted how these advanced strategies collectively offer concrete ways to overcome the pharmacokinetic and formulation challenges posed by ligustilide, and discussed the suitability of these approaches for future preclinical or clinical development.
We believe these additions significantly strengthen the manuscript by offering a forward-looking perspective on how ligand-delivery problems may be overcome, thereby addressing the reviewer’s concern about the lack of discussion of recent drug-delivery strategies. We hope this revision meets your expectations and adds real value to the paper. The following references have been added.
- Yan, R.; Ko, N.L.; Li, S.L.; Tam, Y.K.; Lin, G. Pharmacokinetics and metabolism of ligustilide, a major bioactive component in Rhizoma Chuanxiong, in the rat. Drug Metab Dispos 2008, 36, 400–408, doi:10.1124/dmd.107.017707.
- Zhang, Y.; Zhang, Y.; Han, Y.; Tian, Y.; Wu, P.; Xin, A.; Wei, X.; Shi, Y.; Zhang, Z.; Su, G.; et al. Pharmacokinetics, tissue distribution, and safety evaluation of a ligustilide derivative (LIGc). J Pharm Biomed Anal 2020, 182, 113140, doi:10.1016/j.jpba.2020.113140.
- Ma, Z.; Bai, L. Anti-inflammatory effects of Z-ligustilide nanoemulsion. Inflammation 2013, 36, 294–299, doi:10.1007/s10753-012-9546-2.
- Ke, G.; Zhang, M.; Hu, P.; Zhang, J.; Naeem, A.; Wang, L.; Xu, H.; Liu, Y.; Cao, M.; Zheng, Q. Exploratory Study on Nanoparticle Co-Delivery of Temozolomide and Ligustilide for Enhanced Brain Tumor Therapy. Pharmaceutics 2025, 17, 191.
- Zhang, J.; Xu, W.; Meng, F.; Yi, T. A Spray-Dried Self-Stabilizing Nanocrystal Emulsion of Traditional Chinese Medicine: Preparation, Characterization and ex vivo Intestinal Absorption. Pharmaceutical Fronts 2024, 06, e449–e458, doi:10.1055/s-0044-1791831.
Comment #2
While mechanisms are described, some sections (e.g., Z-Ligustilide’s effect on cisplatin resistance in lung cancer) lack detailed molecular explanations. I would suggest where possible, provide more mechanistic detail or acknowledge gaps in current understanding.
Response
We appreciate the reviewer’s insightful and constructive suggestion. In response, we have substantially revised the section discussing (Z)-Ligustilide and its effects on cisplatin-resistant NSCLC cell lines to enhance the mechanistic depth and clarity. As the original study by Geng et al. did not examine the molecular mechanisms responsible for the observed reduction in cisplatin resistance, we have supplemented this section with additional background describing key pathways known to contribute to cisplatin resistance in NSCLC, including the dysregulation of drug-efflux transporters, enhanced DNA-damage repair capacity, and activation of survival signaling pathways such as PI3K/Akt and MAPK. We also explicitly state that, despite these established mechanisms, the specific molecular targets or pathways modulated by (Z)-Ligustilide remain undefined. By incorporating this contextual information and acknowledging the existing gap in mechanistic understanding, we have more clearly positioned the findings within the broader landscape of cisplatin resistance research. These revisions have been added to Lines 689–693 on Pages 20-21 and Lines 703–706 on Page 21 of the revised manuscript.
Comment #3
The manuscript references several figures and tables, but some are not fully described in the text, and their legends could be more informative. I would suggest to ensure all figures and tables are clearly explained and directly referenced in the relevant sections.
Response
We appreciate the reviewer’s careful assessment and constructive suggestion. In response, we have thoroughly revised the manuscript to ensure that all figures and tables are clearly introduced, referenced, and described in the corresponding sections. Specifically:
- We added detailed first-mention explanations for the Graphical Abstract (Lines 48-55 on Page 2) and Figure 2 (Lines 372-377 on Page 9), which were previously less elaborated compared with Figures 1, 3, and 4.
- We verified that all figures and tables are now directly cited and contextualized at appropriate points in the text.
These revisions enhance the clarity and coherence of the manuscript and ensure that all visual materials are fully integrated into the narrative.
We thank the reviewer for this helpful recommendation.
Comment #4
The conclusions section outlines areas for future research but could benefit from more concrete recommendations, such as prioritizing specific cancer types for clinical trials or suggesting collaborative research models.
Response
We thank the reviewer for this valuable suggestion. In response, we have revised the Conclusions section to include explicit and concrete recommendations for future research. Specifically, we now:
- Prioritize cancer types—including TAM-resistant breast cancer, prostate cancer, and cisplatin-resistant lung cancer—as leading candidates for early-phase clinical evaluation based on the strongest preclinical and mechanistic evidence.
- Propose collaborative research models, such as multi-institutional consortia integrating molecular oncology, pharmacology, and medicinal chemistry, as well as academic–industry partnerships to accelerate formulation development and biomarker-guided studies.
A new paragraph incorporating these recommendations has been added to the Conclusions section (Lines 943-956 on Page 27) in the revised manuscript. We believe this addition enhances the clarity, translational relevance, and practical applicability of the manuscript’s final section.
Comment #5
The manuscript is generally well-written, but minor grammatical and typographical errors are present. A careful proofreading would improve readability.
Response
We sincerely thank the reviewer for noting the presence of minor grammatical and typographical errors in the manuscript. In response, we have conducted a thorough and careful proofreading of the entire document. All necessary English language modifications—including corrections to grammar, spelling, punctuation, and phrasing—have been implemented throughout the manuscript. To facilitate the reviewer’s evaluation of these revisions, all changes have been clearly highlighted in green. We hope that these improvements enhance the overall clarity and readability of the manuscript.
Comment #6
Abbreviations: The list of abbreviations is comprehensive, but some terms could be defined upon first use in the main text for clarity.
Response
We appreciate the reviewer’s observation regarding the use of abbreviations. The journal’s guidelines allow authors to include a comprehensive list of abbreviations, and our manuscript follows this format to ensure consistency and avoid unnecessary repetition within the main text. Since all abbreviations used throughout the manuscript are clearly defined in the dedicated Abbreviations List, readers can easily refer to this section whenever needed.
To maintain coherence with the journal’s permitted structure and to prevent interrupting the flow of the text with repeated definitions, we kindly request to keep the abbreviations defined exclusively in the Abbreviations List. We believe this approach preserves clarity while ensuring the manuscript remains concise and aligned with the journal’s formatting standards.
I, the corresponding author of the manuscript "Ligustilide: A Phytochemical with Potential in Combating Cancer Development and Progression — A Comprehensive and Critical Review" under the assigned ID pharmaceuticals-3976372, on behalf of my coauthors, once again extend my heartfelt gratitude to the knowledgeable Editor-in-Chief and reviewers for their time and expertise in revising our manuscript. After we addressed their constructive and refined feedback and suggestions, a significantly improved manuscript version emerged. Undoubtedly, their insightful suggestions and feedback have significantly enhanced the quality of our manuscript. We respectfully are at the disposal of the Editor-in-Chief and the Reviewer to address any additional suggestions regarding our publication. Suppose you are satisfied with our newly refined and significantly improved version. In that case, we look forward to the acceptance of our article for publication in the prestigious journal Pharmaceuticals. Thank you once again for your time and expertise.
Reviewer 2 Report
Comments and Suggestions for Authors
The authors of the article provide an overview of the pharmacological properties of ligustilide and its cis-isomer (Z)-Ligustilide, paying attention to the anticancer potential: inhibition of proliferation, induction of apoptosis, modulation of autophagy, induction of ferroptosis, increased sensitivity to chemotherapy drugs and epigenetic regulation. The authors note encouraging preclinical results, but emphasize the lack of data on molecular mechanisms, FC/DB, and bioavailability. After a thorough study of this manuscript, I want to summarize the advantages of the work more (Current topic: cancer and drug resistance are a significant clinical problem, a comprehensive coverage of the mechanisms of action (apoptosis, autophagy, ferroptosis, epigenetics), mentioning the prospects of combinations with chemotherapy and personalized therapy is a useful direction for translation into the clinic), rather than the disadvantages. Therefore, after making the changes, I recommend this review in pharmaceuticals.
My recommendations for the review edits.
1. In my opinion, it's necessary to clearly separate the levels of evidence: in vitro, in vivo (rodents), preclinical models of resistance and clinical data (if available). This is sometimes mixed up in the text, which creates a false impression of clinical readiness. It is recommended to add a table listing the key studies: cancer type, model (cellular/animal), dose/method of administration, main endpoints and results.
2. The descriptions of the mechanisms look fragmentary. It is necessary to expand the signaling pathways more deeply (for example, PI3K/AKT, MAPK, p53, NF-kB, Nrf2, factors related to the regulation of autophagy and ferroptosis) and provide direct evidence (citation of works with inhibitors/sensitization, knockdown/overexpression). It is advisable to critically evaluate which mechanisms are confirmed by several independent groups, and which are based on single observations.
3. The authors rightly point out the lack of data. It is necessary to expand the FC/DB section: metabolism (liver enzymes), plasma stability, Cmax, Tmax, T1/2, tissue distribution (especially for solid tumors and in neuroprotection), possible metabolites and their activity. I recommend including an overview of methods for increasing bioavailability (formulations: nanocarriers, liposomes, propellants), as well as the risks associated with rapid metabolism.
4. The article pays little attention to the profile of toxicity and tolerance — MTD, organ toxicity, genotoxicity, cardiotoxicity and possible drug interactions. A review of the available data and a proposal for necessary preclinical studies are required.
5. It seems to me that claims about overcoming resistance should be supported by rigorous experiments: combined in vitro/in vivo trials with the calculation of synergy indices (for example, CI according to Chou-Talalay), analysis of mechanisms (expression of drug transporters, apoptosis/autophagy, epigenetic modifications).
6. I consider it necessary to insert specific recommendations on the stages of translation: necessary preclinical studies before phase I, proposed biomarkers of response, potential combinations with existing drugs, clinical trial designs.
Author Response
RESPONSE TO REVIEWERS' COMMENTS
Manuscript number: pharmaceuticals-3976372 ― Pharmaceuticals (MDPI)
"Ligustilide: A Phytochemical with Potential in Combating Cancer Development and Progression — A Comprehensive and Critical Review"
The authors of this document wish to express their deepest gratitude to the Editor-in-Chief and the Reviewer for their thorough and insightful evaluation of our manuscript. Their expert feedback has been invaluable in enhancing the quality of our work. We have carefully considered and diligently implemented each suggestion, which has significantly improved the manuscript. We have made substantial revisions to address the points raised. These noteworthy changes are marked mainly with YELLOW-highlighted text throughout the document for ease of reference. A note will be provided for the referee's attention, highlighting corrections in a different color. Additionally, we have prepared a detailed and comprehensive response to each comment and suggestion. This response is organized in a "point-by-point" format below, ensuring that every concern has been thoroughly addressed and explained. We sincerely appreciate the time and effort invested by the Editor-in-Chief and the Reviewer, and we believe their contributions have significantly strengthened the final version of our manuscript.
REVIEWER #2
General comment
The authors of the article provide an overview of the pharmacological properties of ligustilide and its cis-isomer (Z)-Ligustilide, paying attention to the anticancer potential: inhibition of proliferation, induction of apoptosis, modulation of autophagy, induction of ferroptosis, increased sensitivity to chemotherapy drugs and epigenetic regulation. The authors note encouraging preclinical results, but emphasize the lack of data on molecular mechanisms, FC/DB, and bioavailability. After a thorough study of this manuscript, I want to summarize the advantages of the work more (Current topic: cancer and drug resistance are a significant clinical problem, a comprehensive coverage of the mechanisms of action (apoptosis, autophagy, ferroptosis, epigenetics), mentioning the prospects of combinations with chemotherapy and personalized therapy is a useful direction for translation into the clinic), rather than the disadvantages. Therefore, after making the changes, I recommend this review in pharmaceuticals.
General response
Dear Erudite Reviewer, thank you for taking the time to revise our manuscript and allowing us to improve based on your valuable comments and suggestions. After addressing all your comments and suggestions regarding our manuscript text, we are confident that a significantly enhanced manuscript version has emerged. We are excited to resubmit the modified version for your perusal and reevaluation. Thank you for your brilliant insights, essential contributions, and feedback. You do have an eye for improvement. As a gesture of our utmost respect for you, we would like to provide you with a detailed and comprehensive point-by-point response to your comments below. Thank you once again for your time and patience in revising our article.
Comment #1
In my opinion, it's necessary to clearly separate the levels of evidence: in vitro, in vivo (rodents), preclinical models of resistance and clinical data (if available). This is sometimes mixed up in the text, which creates a false impression of clinical readiness. It is recommended to add a table listing the key studies: cancer type, model (cellular/animal), dose/method of administration, main endpoints and results.
Response
We thank the reviewer for this valuable suggestion. In response, we have added a new summary table (Table 3) that clearly distinguishes the evidence levels (in vitro, in vivo, and preclinical rodent models) for both ligustilide and (Z)-ligustilide across all cancer types evaluated in the literature.
To avoid redundancy with the detailed mechanistic Tables 1 and 2, Table 3 presents only the model type, model used, dose/exposure, primary outcome, and corresponding reference, providing a concise overview of translational relevance without repeating previously presented mechanistic information.
This new table greatly improves clarity by enabling readers to quickly identify the type of experimental model, the general findings, and the strength of evidence supporting each anticancer effect.
Please find the modifications on Pages 22-23 of the revised manuscript document.
We believe this revision fully addresses the reviewer’s concern and enhances the manuscript's readability.
Comment #2
The descriptions of the mechanisms look fragmentary. It is necessary to expand the signaling pathways more deeply (for example, PI3K/AKT, MAPK, p53, NF-kB, Nrf2, factors related to the regulation of autophagy and ferroptosis) and provide direct evidence (citation of works with inhibitors/sensitization, knockdown/overexpression). It is advisable to critically evaluate which mechanisms are confirmed by several independent groups, and which are based on single observations.
Response
We appreciate the reviewer’s thoughtful suggestion. In response, we expanded the mechanistic interpretation in the manuscript and now provide a clearer, more integrative synthesis of the pathways influenced by Ligustilide and (Z)-Ligustilide across the included studies. The revised text highlights how these compounds modulate key oncogenic signaling cascades, including PI3K/Akt, MAPK pathways (ERK, JNK, p38), p53-mediated apoptotic signaling, the Nrf2/HO-1 oxidative-stress response, and regulators of autophagy.
We also describe additional mechanisms that appear to be cancer-type specific, such as ferroptosis in acute myeloid leukemia and ER-stress–associated radiosensitization in oral cancer. To strengthen the scientific interpretation, we now comment on the consistency of these mechanisms across studies and note that most available evidence is correlative, with limited use of pathway-specific inhibitors or genetic approaches. This provides a more critical and comprehensive analysis, as requested.
The modifications can be found on Lines 957-979 of Pages 27-28 of the revised manuscript.
We thank the reviewer for this valuable recommendation, which has helped us substantially improve the manuscript's mechanistic depth and clarity.
Comment #3
The authors rightly point out the lack of data. It is necessary to expand the FC/DB section: metabolism (liver enzymes), plasma stability, Cmax, Tmax, T1/2, tissue distribution (especially for solid tumors and in neuroprotection), possible metabolites and their activity. I recommend including an overview of methods for increasing bioavailability (formulations: nanocarriers, liposomes, propellants), as well as the risks associated with rapid metabolism.
Response
We thank the reviewer for this insightful and constructive recommendation. In accordance with the suggestions, we have substantially expanded the pharmacokinetic and biopharmaceutical (FC/DB) discussion of Ligustilide. Specifically, we have:
- Added detailed information on metabolic pathways, including the involvement of hepatic CYP enzymes and characterization of primary metabolic routes (epoxidation, hydroxylation, aromatization, and glutathionylation).
- Discussed plasma stability and the rapid hepatic clearance that contributes to poor systemic exposure.
- Expanded the section on tissue distribution, including considerations relevant to neuroprotection and potential limitations regarding distribution to solid tumors.
- Provided additional detail on primary Ligustilide metabolites, such as senkyunolide I and senkyunolide H, and briefly addressed their known biological activities.
- Added a new subsection reviewing strategies to enhance bioavailability, including nanoformulations, liposomes, nanoemulsions, solid lipid nanoparticles, and propellant-based systems.
- Discussed the risks associated with rapid metabolism, including limited therapeutic exposure and the potential formation of reactive intermediates.
The main additions are now included in the manuscript as a new subsection in Lines 266-301 on Page 7 of the revised manuscript document.
The section that was newly added is named “Additional Considerations: Metabolism, Plasma Stability, Tissue Distribution, and Strategies to Improve Bioavailability.”
We believe these revisions significantly strengthen the pharmacokinetic component of the review and fully address the reviewer’s comment.
Comment #4
The article pays little attention to the profile of toxicity and tolerance — MTD, organ toxicity, genotoxicity, cardiotoxicity and possible drug interactions. A review of the available data and a proposal for necessary preclinical studies are required.
Response
We thank the reviewer for this critical point. We agree that the toxicological characterization of Ligustilide is essential for its further development. However, published data on the isolated compound remain limited. In the revised manuscript, we now (i) summarize all available information on acute and subacute toxicity, cytotoxicity, potential organ-specific effects, and reported safety concerns derived from studies using purified Ligustilide or Ligustilide-rich extracts, and (ii) highlight the significant knowledge gaps.
Because no formal MTD, genotoxicity, cardiotoxicity, or comprehensive drug–drug interaction studies have been reported in the literature, we now propose a set of necessary preclinical studies, including dose-range finding and MTD determination, safety pharmacology (cardiovascular, neurological, respiratory), standard genotoxicity assays (Ames, micronucleus), subchronic toxicity in two species, and CYP-mediated interaction profiling.
The modifications can be found on Lines 230-255 of Pages 6-7 of the revised manuscript.
Comment #5
It seems to me that claims about overcoming resistance should be supported by rigorous experiments: combined in vitro/in vivo trials with the calculation of synergy indices (for example, CI according to Chou-Talalay), analysis of mechanisms (expression of drug transporters, apoptosis/autophagy, epigenetic modifications).
Response
We thank the reviewer for this critical and constructive comment. We agree that claims regarding the ability of (Z)-Ligustilide to overcome therapeutic resistance must be supported by rigorous experimental methodologies, including synergy analysis, combined in vitro/in vivo validation, and mechanistic assays of resistance pathways.
To address this, we have revised the Conclusions section to clearly acknowledge that the existing evidence for resistance reversal is preliminary and primarily based on in vitro findings and molecular markers of restored sensitivity. We have also clarified that most published studies did not include formal synergy index calculations, nor systematic mechanistic validation involving drug-efflux transporters, apoptosis/autophagy rescue assays, or epigenetic reversibility analyses.
We have now added the text in Lines 921-931 on Pages 26-27 to the manuscript:
These additions clarify the limitations of the current literature and align our conclusions with the level of experimental rigor highlighted by the reviewer. We believe these revisions strengthen the manuscript and accurately frame the current state of evidence.
Comment #6
I consider it necessary to insert specific recommendations on the stages of translation: necessary preclinical studies before phase I, proposed biomarkers of response, potential combinations with existing drugs, clinical trial designs.
Response
We thank the reviewer for this valuable suggestion. We agree that outlining explicit steps for translational development will significantly strengthen the manuscript. In response, we have added a dedicated text within the “Recommendations for Clinical Translation” portion, providing specific recommendations for (i) essential preclinical studies before phase I trials, (ii) candidate biomarkers of therapeutic response, (iii) rational drug combinations, and (iv) potential early-phase clinical trial designs.
These additions now provide a structured roadmap for the clinical translation of Ligustilide and (Z)-Ligustilide and can be found in Lines 808-854 on Pages 24-25 of the revised manuscript document.
I, the corresponding author of the manuscript "Ligustilide: A Phytochemical with Potential in Combating Cancer Development and Progression — A Comprehensive and Critical Review" under the assigned ID pharmaceuticals-3976372, on behalf of my coauthors, once again extend my heartfelt gratitude to the knowledgeable Editor-in-Chief and reviewers for their time and expertise in revising our manuscript. After we addressed their constructive and refined feedback and suggestions, a significantly improved manuscript version emerged. Undoubtedly, their insightful suggestions and feedback have significantly enhanced the quality of our manuscript. We respectfully are at the disposal of the Editor-in-Chief and the Reviewer to address any additional suggestions regarding our publication. Suppose you are satisfied with our newly refined and significantly improved version. In that case, we look forward to the acceptance of our article for publication in the prestigious journal Pharmaceuticals. Thank you once again for your time and expertise.
Reviewer 3 Report
Comments and Suggestions for Authors
Comments and Suggestions for Authors
The review article titled “Ligustilide: A Phytochemical with Potential in Combating Cancer Development and Progression — A Comprehensive and Critical Review, this review article is interesting. The authors have written well and covered the key topics that need to be reviewed in accordance with the title. However, there are some points that should be improved to make the article more engaging and easier to understand, as follows:
Remarks:
- Page 4/30, Line 118-135
- Ligustilide: Unveiling Its Biosynthesis, Physicochemical Properties, and Pharmacokinetics
### The authors should review the sources where this compound has been identified and summarize the findings, including the plant name, plant part, and the amount detected. Presenting this information in a table would make it easier to understand.
- Page 8-9/30, Line 317-363
- Ligustilide in Cancer Prevention and Intervention
4.1. Literature Search Methodology
….Keywords were used to facilitate the literature review. They included terms such as “Ligustilide,” “cancer cell lines,” “animal models,” “cancer,” “cell lines,” “breast cancer,” “oral cancer,” “gastric cancer,” “lung cancer,” and “signaling pathways,” alongside biological processes like “apoptosis,” “cell proliferation,” “metastasis,” “cell death,” “cell cycle,” “PI3K,” “Nrf2,” “mTOR,” and “NF-kB.”…..
### Since the title refers to cancer prevention, it is unclear which keywords the authors used to search for relevant studies. The keywords listed seem to be related to cancer treatment rather than cancer prevention.
- Page 9-10/30, Line 364-376
4.2. Literature Search Report: Results of Literature Search Following PRISMA Guidelines
and Figure 3 Report of the Included Studies following the PRISMA Guidelines [42]
### The authors should clearly specify the exclusion criteria at each stage. For example, in Figure 3, in the section labeled “Records excluded (n = 64),” please indicate the criteria used for exclusion.
### I think that in the section stating “Reports excluded: Not English language (n = 4); Not involving Ligustilide (n = 4); Non-experimental paper (n = 2),” the criterion “Not English language (n = 4)” should have been applied from the initial screening step. Why did the authors use this criterion only in the later stage of the selection process?
- The authors should provide a summary indicating whether Ligustilide or (Z)-Ligustilide has a greater effect on specific types of cancer. Since the manuscript discusses the effects of these compounds on various cancer types, readers may find it difficult to determine which form of the compound is more effective against which type of cancer.
- The authors should also review the toxicity of the compound toward normal cells, as including this aspect would make the article more informative and interesting.
Cheers,
Date of this review
9 November 2025
Author Response
RESPONSE TO REVIEWERS' COMMENTS
Manuscript number: pharmaceuticals-3976372 ― Pharmaceuticals (MDPI)
"Ligustilide: A Phytochemical with Potential in Combating Cancer Development and Progression — A Comprehensive and Critical Review"
The authors of this document wish to express their deepest gratitude to the Editor-in-Chief and the Reviewer for their thorough and insightful evaluation of our manuscript. Their expert feedback has been invaluable in enhancing the quality of our work. We have carefully considered and diligently implemented each suggestion, which has significantly improved the manuscript. We have made substantial revisions to address the points raised. These noteworthy changes are marked mainly with YELLOW-highlighted text throughout the document for ease of reference. A note will be provided for the referee's attention, highlighting corrections in a different color. Additionally, we have prepared a detailed and comprehensive response to each comment and suggestion. This response is organized in a "point-by-point" format below, ensuring that every concern has been thoroughly addressed and explained. We sincerely appreciate the time and effort invested by the Editor-in-Chief and the Reviewer, and we believe their contributions have significantly strengthened the final version of our manuscript.
REVIEWER #3
General comment
The review article titled “Ligustilide: A Phytochemical with Potential in Combating Cancer Development and Progression — A Comprehensive and Critical Review, this review article is interesting. The authors have written well and covered the key topics that need to be reviewed in accordance with the title. However, there are some points that should be improved to make the article more engaging and easier to understand, as follows.
General response
Dear Erudite Reviewer, thank you for taking the time to revise our manuscript and allowing us to improve based on your valuable comments and suggestions. After addressing all your comments and suggestions regarding our manuscript text, we are confident that a significantly enhanced manuscript version has emerged. We are excited to resubmit the modified version for your perusal and reevaluation. Thank you for your brilliant insights, essential contributions, and feedback. You do have an eye for improvement. As a gesture of our utmost respect for you, we would like to provide you with a detailed and comprehensive point-by-point response to your comments below. Thank you once again for your time and patience in revising our article.
Comment #1
Page 4/30, Line 118-135
Ligustilide: Unveiling Its Biosynthesis, Physicochemical Properties, and Pharmacokinetics
### The authors should review the sources where this compound has been identified and summarize the findings, including the plant name, plant part, and the amount detected. Presenting this information in a table would make it easier to understand.
Response
Thank you very much for your comment. We have, in fact, already addressed the concern about variability in Ligustilide content in the manuscript (see Lines 127-142 on Page 4 of the revised manuscript document). In those lines, we report quantitative data showing that the total content of Z- and E-Ligustilide in Angelica sinensis (“danggui”) ranges from 5.63 to 24.53 mg/g, with a mean value of 11.02 mg/g, and explicitly note that these levels differ by cultivation region (13.90 mg/g in Yunnan, 12.51 mg/g in Sichuan, and 10.04 mg/g in Gansu) as shown by Li et al. We further describe that the Ligustilide content was significantly higher in samples with certain morphological traits — namely, small primary roots, long rootlets, and strongly perfumed herbs — which directly addresses your point about variation by herb shape. In addition, the manuscript includes data on other species: Angelica acutiloba (1.00 mg/g) and Levisticum officinale (lovage root, 2.78 mg/g), highlighting apparent inter-species differences. By presenting this information, we not only demonstrate substantial variability in Ligustilide content across species, plant parts, and geographical origins, but also underscore the importance of standardized sourcing and analytical methods in pharmacokinetic and pharmacological studies. Taken together, we believe that the current text fully addresses your concern.
Comment #2
Page 8-9/30, Line 317-363
Ligustilide in Cancer Prevention and Intervention
4.1. Literature Search Methodology
….Keywords were used to facilitate the literature review. They included terms such as “Ligustilide,” “cancer cell lines,” “animal models,” “cancer,” “cell lines,” “breast cancer,” “oral cancer,” “gastric cancer,” “lung cancer,” and “signaling pathways,” alongside biological processes like “apoptosis,” “cell proliferation,” “metastasis,” “cell death,” “cell cycle,” “PI3K,” “Nrf2,” “mTOR,” and “NF-kB.”…..
### Since the title refers to cancer prevention, it is unclear which keywords the authors used to search for relevant studies. The keywords listed seem to be related to cancer treatment rather than cancer prevention.
Response
We appreciate the reviewer’s insightful observation. We agree that the previously listed keywords primarily reflected studies on cancer treatment rather than prevention. To address this concern and ensure our literature search strategy accurately captures research on cancer prevention, we have revised and expanded the keyword set.
In addition to the original terms, we have now incorporated the following prevention-focused keywords: “cancer prevention,” “chemoprevention,” “carcinogenesis prevention,” “DNA damage protection,” and “anti-inflammatory mechanisms.” These terms were selected to align with the preventive aspects of Ligustilide’s biological activities, including its antioxidant, anti-inflammatory, and antimutagenic properties, which are relevant to early-stage cancer inhibition.
These new keywords have been added to the Literature Search Methodology section (Page 10, Lines 408-410) to clearly indicate that our search included both preventive and therapeutic dimensions of Ligustilide research. This revision ensures that our methodology is transparent and appropriately reflects the scope suggested by the manuscript title.
Comment #3
Page 9-10/30, Line 364-376
4.2. Literature Search Report: Results of Literature Search Following PRISMA Guidelines
and Figure 3 Report of the Included Studies following the PRISMA Guidelines [42]
### The authors should clearly specify the exclusion criteria at each stage. For example, in Figure 3, in the section labeled “Records excluded (n = 64),” please indicate the criteria used for exclusion.
### I think that in the section stating “Reports excluded: Not English language (n = 4); Not involving Ligustilide (n = 4); Non-experimental paper (n = 2),” the criterion “Not English language (n = 4)” should have been applied from the initial screening step. Why did the authors use this criterion only in the later stage of the selection process?
Response
We sincerely thank the reviewer for these helpful observations. In response, we have made several revisions to clearly document the study selection process in accordance with PRISMA guidelines.
We have updated Section 4.2 (“Literature Search Report”) and revised the PRISMA flow diagram (Figure 3) to list all exclusion criteria applied explicitly:
- Title and abstract screening,
- Full-text assessment, and
- Final inclusion.
In the revised version, each stage now specifies the precise criteria used and the number of records excluded under each criterion. For example, in the “Records excluded (n = 64)” category, we now detail exclusions due to insufficient methodological detail, studies focusing solely on pharmacokinetics/toxicokinetics, duplicates, and in vitro or in vivo studies lacking outcome variables of interest.
We appreciate the reviewer’s point that non-English articles are typically excluded during the initial screening stage. In our workflow, however, the language of several records could not be confidently determined from the title and abstract alone, as some entries were indexed with English titles but did not indicate the language of the full text. These records were only identified as non-English when full-text retrieval was attempted. Therefore, although the language restriction was part of our predefined eligibility criteria, these exclusions necessarily occurred at the full-text assessment stage.
To prevent any ambiguity, we have now added a clarification in Section 4.2 stating:
“Although articles were intended to be limited to English-language publications, the language of several records was not clearly identifiable during title/abstract screening. These records were excluded during the full-text assessment phase once it became evident that the full text was not available in English.” (Lines 448-451 on Page 11)
The PRISMA flow diagram (Figure 3 on Page 12) has been revised to:
- Reflect the exact number of studies excluded at each stage and,
- Include detailed exclusion reasons within each stage.
We believe these revisions make the selection process more transparent and fully aligned with the reviewer’s recommendation.
Comment #4
The authors should provide a summary indicating whether Ligustilide or (Z)-Ligustilide has a greater effect on specific types of cancer. Since the manuscript discusses the effects of these compounds on various cancer types, readers may find it difficult to determine which form of the compound is more effective against which type of cancer.
Response
We sincerely thank the reviewer for this valuable suggestion. We agree that, given the breadth of cancer types discussed in this manuscript, a consolidated comparison of Ligustilide versus (Z)-Ligustilide would greatly enhance clarity and help readers quickly understand which compound demonstrates more substantial or more specific anticancer activity.
In response, we have substantially revised the Conclusions section to include a comprehensive summary that synthesizes the comparative anticancer effects of both compounds across all cancer models evaluated in the manuscript. This new text explicitly highlights:
- Which cancers show stronger or more consistent responses to Ligustilide, emphasizing its broad-spectrum activity and well-characterized mechanisms across gastric, bile duct, bladder, hepatocellular, osteoblastoma, and lung cancers.
- Which cancers demonstrate more potent or specialized responses to (Z)-Ligustilide, particularly its unique therapeutic value in drug-resistant or epigenetically dysregulated cancers such as tamoxifen-resistant breast cancer, AML, oral cancer, ovarian cancer, and glioblastoma.
- The mechanistic distinctions between the two compounds, including Ligustilide’s broad apoptotic, autophagic, metabolic, and microenvironment-modulating effects compared with (Z)-Ligustilide’s targeted influence on epigenetic regulation, ferroptosis, drug-resistance reversal, redox modulation, and ER stress pathways.
- An overall interpretation indicating that Ligustilide exhibits wider general anticancer efficacy across numerous tumor types. At the same time, (Z)-Ligustilide offers more specialized therapeutic advantages in cancers characterized by therapeutic resistance or aggressive phenotypes.
This expanded comparative synthesis in the Conclusions section directly addresses the reviewer’s concern by providing readers with a clear, concise, and integrated understanding of the relative strengths and therapeutic potential of each compound. We believe this revision significantly enhances the manuscript’s clarity and impact. The major modifications are found in Lines 862-871 on Page 25 of the revised manuscript document.
Thank you again for helping us improve the quality of the work.
Comment #5
The authors should also review the toxicity of the compound toward normal cells, as including this aspect would make the article more informative and interesting.
Response
We thank the reviewer for this critical point. We agree that the toxicological characterization of Ligustilide is essential for its further development. However, published data on the isolated compound remain limited. In the revised manuscript, we now (i) summarize all available information on acute and subacute toxicity, cytotoxicity, potential organ-specific effects, and reported safety concerns derived from studies using purified Ligustilide or Ligustilide-rich extracts, and (ii) highlight the significant knowledge gaps.
Because no formal MTD, genotoxicity, cardiotoxicity, or comprehensive drug–drug interaction studies have been reported in the literature, we now propose a set of necessary preclinical studies, including dose-range finding and MTD determination, safety pharmacology (cardiovascular, neurological, respiratory), standard genotoxicity assays (Ames, micronucleus), subchronic toxicity in two species, and CYP-mediated interaction profiling.
The modifications can be found on Lines 230-255 of Pages 6-7 of the revised manuscript.
I, the corresponding author of the manuscript "Ligustilide: A Phytochemical with Potential in Combating Cancer Development and Progression — A Comprehensive and Critical Review" under the assigned ID pharmaceuticals-3976372, on behalf of my coauthors, once again extend my heartfelt gratitude to the knowledgeable Editor-in-Chief and reviewers for their time and expertise in revising our manuscript. After we addressed their constructive and refined feedback and suggestions, a significantly improved manuscript version emerged. Undoubtedly, their insightful suggestions and feedback have significantly enhanced the quality of our manuscript. We respectfully are at the disposal of the Editor-in-Chief and the Reviewer to address any additional suggestions regarding our publication. Suppose you are satisfied with our newly refined and significantly improved version. In that case, we look forward to the acceptance of our article for publication in the prestigious journal Pharmaceuticals. Thank you once again for your time and expertise.